# RankSHAP: Shapley Value Based Feature Attributions for Learning to Rank

**Tanya Chowdhury, Yair Zick, James Allan**
Manning College of Information and Computer Sciences
University of Massachusetts Amherst
{tchowdhury,yzick,allan}@cs.umass.edu

## Abstract

Numerous works propose post-hoc, model-agnostic explanations for learning to rank, focusing on ordering entities by their relevance to a query through feature attribution methods. However, these attributions often weakly correlate or contradict each other, confusing end users. We adopt an axiomatic game-theoretic approach, popular in the feature attribution community, to identify a set of fundamental axioms that every ranking-based feature attribution method should satisfy. We then introduce Rank-SHAP, extending classical Shapley values to ranking. We evaluate the RankSHAP framework through extensive experiments on two datasets, multiple ranking methods and evaluation metrics. Additionally, a user study confirms RankSHAP's alignment with human intuition. We also perform an axiomatic analysis of existing rank attribution algorithms to determine their compliance with our proposed axioms. Ultimately, our aim is to equip practitioners with a set of axiomatically backed feature attribution methods for studying IR ranking models, that ensure generality as well as consistency.

## 1 Introduction

A user submits a query to a search engine, wanting to know, say, what is the best car to purchase. The search engine processes the query and outputs a ranked list of documents (web pages) in response to the query. How did the search engine decide on the order in which the documents are presented? The task of ranking entails a query $\vec{q}$ and a set of documents $D$ as input to a ranking model $f_R$, which in turn generates an ordering of these documents. We assume that the ranking model is available only as a black-box which can be queried using an API. For a particular query, we wish to understand the logic behind the generated ordering, to verify the functionality of the model as well as to help end users. Ranking feature attributions are critical for building trust in systems which are used for high stakes decision making (Anand et al., 2022; Chowdhury et al., 2023). Such feature attributions are also useful in ruling out prejudices in widely used web search, product search and recommendation systems (Singh & Anand, 2019; Singh et al., 2020).

Additive feature attributions have become a well studied topic in recent years, especially for regression and classification tasks. To maintain a consistency between different feature attribution methods, various authors (Lundberg & Lee, 2017; Datta et al., 2016) proposed a unique solution based on the Shapley value (Shapley, 1953). This solution is considered the reference standard of feature attributions because of the several fundamental axioms it uniquely satisfies. Different approximations such as KernelSHAP (Lundberg & Lee, 2017) and DeepSHAP (Lundberg & Lee, 2017; Shrikumar et al., 2017) are used for popular real-world applications.

Quite a few attempts have been made to generate feature attributions for ranking tasks, often by extending attribution methods from classification/regression. EXS (Singh & Anand, 2019) and Rank-LIME (Chowdhury et al., 2023) extend LIME in different manners to obtain ranking feature attributions. However, these attributions contradict each other across the different settings in the very same algorithm. Fernando et al. (2019) also generate explanations for ranking models using DeepSHAP and report little correlation between the attributions generated by different *reference values* for the very same query. This shows us that extensions of popular feature attribution methods for classification and regression are not good attribution candidates for ranking. More recently,

RankingSHAP (Heuss et al., 2024) extend Shapley values to generate feature attributions for ordered lists. However, we show that even their proposed-framework does not satisfy fundamental properties like *monotonicity* and *efficiency*. Consequently, establishing a robust reference standard for ranking feature attributions is imperative. Such a framework will significantly benefit a wide range of stakeholders, from end users seeking more reliable search results to researchers and practitioners aiming to gain deeper insights into model behavior.

**Our Contributions.** We present a set of fundamental axioms for Information Retrieval (IR) value functions—*Relevance Sensitivity* and *Position Sensitivity*—along with Shapley properties specific to ranking: *Rank-Efficiency*, *Rank-Missingness*, *Rank-Symmetry*, and *Rank-Monotonicity*. We believe, these axioms are essential for any ranking feature attribution method to uphold without sacrificing generality. To address this need, we introduce *RankSHAP*, an extension of the Shapley value that satisfies all the aforementioned axioms. We discuss computationally feasible approximations using KernelSHAP to make RankSHAP practical for real-world applications. We conduct extensive experiments on two datasets—MS MARCO and Robust04—using multiple document re-ranking models based on BERT, T5, and LLAMA2. Our results demonstrate that RankSHAP outperforms existing ranking feature attribution methods, surpassing the best baseline by $25.78\%$ in *Fidelity* and $19.68\%$ in *weighted Fidelity* (wFidelity). Additionally, we carry out an IRB-approved user study to assess how well the proposed axioms align with human intuition. We also provide an axiomatic analysis of existing rank attribution methods—EXS, RankLIME, and RankingSHAP—to determine whether they satisfy the fundamental axioms. Ultimately, we advocate for practitioners to adopt RankSHAP-based, axiomatically grounded feature attribution methods as the reference standard for their IR explanation needs. Once the work is accepted, we plan to release our code as a python library.

## 2 BACKGROUND

We start by briefly describing the Shapley value, as used for credit assignment in coalitional game theory, and its consequent use in feature attributions for machine learning models. Following this, we outline some basic properties of information retrieval systems and discuss evaluation metrics for ordered document lists.

### 2.1 BASIC AXIOMS AND THE SHAPLEY VALUE

The Shapley axioms for feature attributions have been borrowed from coalitional game theory, where they are foundational constraints, used to fairly distribute costs/rewards among a set of players. In the context of feature attributions for ML models, we can think of each feature as a *player* contributing to the model's output. Thus, given any model $f$, that has a scalar output, the feature attribution of feature $i$ in the decision made by $f$ towards the model input $\vec{s}$ is represented by $\phi_i(f, \vec{s})$. Let us assume there are a total of $m$ features. For a feature attribution $\phi_i(f, \vec{s})$ to be *sound*, it is necessary for it to satisfy certain basic axioms, namely: Efficiency, Missingness, Symmetry and Monotonicity (Appendix B).

The *Shapley value* (Young, 1985; Shapley, 1953) uniquely satisfies the four axioms above and has been widely adopted in feature attribution for regression/classification tasks. It is defined as:

$$\phi_i(f, \vec{s}) = \sum_{\vec{z} \subseteq \vec{s}} \frac{|\vec{z}|!(m - |\vec{z}| - 1)!}{m!} \left[ f(\vec{z}) - f(\vec{z} \setminus i) \right] \tag{1}$$

where $\vec{z} \subseteq \vec{s}$ represents all binary vectors $\vec{z}$ where the non-zero entries correspond to a subset of the non-zero entries in $\vec{s}$. Essentially, $\vec{z}$ indicates which features are included (non-zero) and which are excluded (zero) in a particular subset; $|\vec{z}|$ is the number of non-zero entries in $\vec{z}$; $\vec{z} \setminus i$ denotes the vector $\vec{z}$ with the $i$-th feature removed (set to zero/replaced by baseline value); $f(\vec{z})$ denotes the output of the model $f$ when the input features corresponding to the non-zero entries in $\vec{z}$ are set to their values in $\vec{s}$, and the remaining features are set to some baseline value (e.g., zero).

Satisfying these basic axioms ensures that a feature attribution method is fair and consistent, and as a result, trustworthy for end users. The classical Shapley framework cannot be directly applied to the ranking feature attribution task as in the ranking scenario, the model output $f_R(\vec{s})$ is not scalar, but an ordering of documents.

## 2.2 DOCUMENT RELEVANCE AND IR EVALUATION METRICS

In Information Retrieval (IR), the concept of *relevance* is fundamental to understanding how search systems operate. Relevance $rel(\vec{q}, \vec{d_i})$ quantifies how well a document $\vec{d_i}$ matches a user's query $\vec{q}$. For simplicity, the relevance score of a particular document $\vec{d_j}$ can be denoted as $rel_j$. These scores can be binary (relevant or not relevant) or numerical values indicating varying degrees of relevance, with higher scores representing better matches. Assigning relevance scores allows IR systems to prioritize the most pertinent documents in search results, enhancing the user's ability to find useful information quickly.

Evaluating the quality of an ordered list of documents (e.g., web pages) is crucial for assessing how effectively an IR system meets users' needs. Given an ordered list, we need to determine how well this arrangement serves the user compared to other possible orderings. Several evaluation metrics are commonly used for this purpose. *Precision* measures the proportion of retrieved documents that are relevant. *Mean Average Precision (MAP)* calculates the average precision across multiple queries, considering the rank positions of all relevant documents. *Discounted Cumulative Gain (DCG)* takes into account both the relevance of documents and their positions in the ranked list. It applies a logarithmic discount factor to penalize relevant documents that appear lower in the list. *Normalized Discounted Cumulative Gain (NDCG)* (Järvelin & Kekäläinen, 2002) is a normalized version of DCG that scales the scores between 0 and 1. This allows for fair comparison across different queries and systems.

Among these metrics, *NDCG* has become the *standard* and is *almost universally used* to evaluate the effectiveness of ranked lists (Yining et al., 2013). Due to its ability to effectively account for both the relevance and the position of documents, *NDCG* is extensively adopted in modern IR systems, search engines, and recommendation platforms to assess and compare the performance of different ranking algorithms (Appendix C).

## 3 THE RANKSHAP FRAMEWORK

We first define the feature attribution problem for ranking models in Section 3.1. We then define the Shapley axioms for the ranking task in Section 3.2. Next, in Section 3.3, we discuss ideal properties of ordered list value functions and propose a Generalized Ranking Effectiveness Metric (*GREM*). Finally, in Section 3.4 we present the unique solution that satisfies the Shapley and *GREM* constraints for the ranking feature attribution task. In Section 3.5, we discuss computationally feasible approximations of the RankSHAP expression, making it suitable for practical use cases.

### 3.1 THE RANKING ATTRIBUTION PROBLEM

Let $f_R$ represent a ranking model, acessible to us only as a black-box. Given a query $\vec{q}$ and a set of documents $D = \{\vec{d_1}, \ldots, \vec{d_k}\}$, $f_R$ computes an ordering of documents in $D$, i.e., $f_R(\vec{q}, D)$. Let the query and set of documents $(\vec{q}, D)$ jointly form the instance $(\vec{x})$. The goal is to generate post-hoc feature attributions $\phi_R(f_R, \vec{x})$, to understand the ranking model's decision corresponding to $\vec{x}$.

We assume that there are $n$ documents in $D$, and that there are $m$ features of interest. $\phi_R(f_R, \vec{x})$ is thus a $m$- dimensional vector where each element represents the contribution of a particular feature to the ranking decision. The query and documents here can be represented in any manner, for example: bag of words, human engineered features, vectors in some embedding space, etc.

### 3.2 SHAPLEY AXIOMS FOR RANKING

Let the set of features being studied be represented by $M = \{1, 2, 3..., m\}$ and let $V_R$ be a function that assigns a real-valued effectiveness score to each ordering of documents. The effectiveness score reflects the quality of the ranking based on the relevance of the documents to the query. We refer to the mean value of a feature across the dataset being studied as its baseline value. Using these, we re-write the Shapley axioms for the ranking task as follows:

**Rank-Efficiency:** The sum total of all feature attributions for a particular ordering should be equal to the difference between the effectiveness scores of the orderings obtained when all features are

included and when all features are replaced by their baseline values.

$$\sum_{i=1}^{m} \phi_R(f_R, \vec{x}, i) = V_R(f_R(\vec{x})) - V_R(f_R(\emptyset))$$

where $f_R(\emptyset)$ represents the model's output when all features are replaced by their baseline values.

**Rank-Missingness:** If the inclusion of a feature $i$ does not change the effectiveness score for any ordering as compared to when the feature is replaced by its baseline value, it must be assigned an attribution of $0$.

More formally, if for all $S \subseteq M \setminus \{i\}$ we have $V_R(f_R(S \cup i, \vec{x})) = V_R(f_R(S, \vec{x}))$, then $\phi_R(f_R, \vec{x}, i) = 0$.

**Rank-Symmetry:** Two features $i, j \in M$ are called *symmetric* if they produce equal effectiveness scores when added individually to every possible feature coalition excluding them. The Rank-symmetry axiom requires that we assign equal attributions to symmetric features.

If for all $S \subseteq M \setminus \{i, j\}$ we have $V_R(f_R(S \cup i, \vec{x})) = V_R(f_R(S \cup j, \vec{x}))$, then $\phi_R(f_R, \vec{x}, i) = \phi_R(f_R, \vec{x}, j)$.

**Rank-Monotonicity:** For input $\vec{x}$ and two ranking models $f_R$ and $f'_R$, if for all coalitions $S$ where $S \subseteq M \setminus \{i\}$, the marginal effectiveness gain of adding $i$ to $S$ in $f_R$, is greater than or equal to the marginal effectiveness gain of adding $i$ to $S$ in $f'_R$, the attribution of $i$ in $f_R$ has to be greater in $f_R$ as compared to $f'_R$.

If for all $S \subseteq M \setminus \{i\}$ we have $V_R(f_R(S \cup i, \vec{x})) - V_R(f_R(S, \vec{x})) \geq V_R(f'_R(S \cup i, \vec{x})) - V_R(f'_R(S, \vec{x}))$ then $\phi_R(f_R, \vec{x}, i) \geq \phi_R(f'_R, \vec{x}, i)$.

### 3.3 A Generalized Ranking Effectiveness Metric (*GREM*)

In this section, we study the desired properties for the ordering effectiveness score $V_R$. The IR evaluation community uses a number of measures to evaluate the effectiveness of a search engine's ranking system. Different measures focus on different aspects of the retrieval and different needs of users (Gupta et al., 2019). A key part of all measures is the relationship between a document's rank and the relevance score: they are expected to be consist with each other. Summarizing most work on IR evaluation leads us to two high-level properties that we expect any metric quanitifying effectiveness of a ranking to provide.

Given a query $\vec{q}$, a set of $n$ documents $D = [\vec{d_1}, \vec{d_2}, \ldots, \vec{d_k}]$, a human-generated relevance value $rel_j$ associated with each document $d_j$, and a metric value $GREM_n$ determined from the set of query-document pairs, consider the ordered list $f_R(\vec{q}, D)$. For an evaluation of the ordering, it is ideal for the metric ($GREM_n$) to satisfy the following axioms:

**Relevance Sensitivity:** If the relevance score of any document increases, while the relevance scores of all other documents remain the same, the metric value $GREM_n$ should not decrease.

Formally, if $rel'_j > rel_j$ for some $j$, and $rel'_i = rel_i$ for all $i \neq j$, then $GREM'_n \geq GREM_n$, where $GREM'_n$ is the metric computed with the updated relevance scores $\{rel'_1, rel'_2, \ldots, rel'_k\}$.

**Position Sensitivity:** If two documents $\vec{d_i}$ and $\vec{d_j}$ are swapped in the ranking, moving the document with the higher relevance score to a higher (better) rank, the metric value $GREM_n$ should not decrease.

*Formally*, let $o$ be the original ordering, and $o'$ be the ordering where only $\vec{d_i}$ and $\vec{d_j}$ are swapped. If $rel_i \geq rel_j$ and $\text{rank}_o(\vec{d_i}) > \text{rank}_o(\vec{d_j})$, then $GREM'_n \geq GREM_n$, where $GREM'_n$ is the metric computed with the new ordering $o'$.

*Relevance Sensitivity* intuitively captures the principle that more relevant documents should enhance the overall effectiveness, while *Position Sensitivity* reflects that users prefer to find relevant information earlier in the ranking. These axioms are fundamental to any rank effectiveness metric because they align with the core goal of retrieval systems: to present the most relevant information prominently, thereby maximizing user satisfaction and system effectiveness.

**Theorem 3.1.** *An ordered list evaluation metric satisfies the axioms of Relevance Sensitivity and Position Sensitivity if and only if it can be represented as*

$$GREM_n = \sum_{j=1}^{n} g(rel_j) \cdot h(j),$$

*where $g(rel_j)$ is a non-decreasing function of the relevance score $rel_j$ (gain function), and $h(j)$ is a non-negative, non-increasing function of the rank position $j$ (discount function).*

Proof of Theorem 3.1 is sketched in Appendix A. Examples of gain functions $g(rel_j)$ include linear gain $rel_j$ and exponential gain $2^{rel_j} - 1$. Examples of discount functions $h(j)$ include no discount $h(j) = 1$, logarithmic discount $h(j) = \frac{1}{\log_2(j+1)}$ and reciprocal discount $h(j) = \frac{1}{j}$. Most widely-used IR metrics like *CG*, *DCG*, *NDCG*, reciprocal rank, precision@k belong to the $GREM_n$ framework.

### 3.4 RANKSHAP ATTRIBUTIONS

Building upon the Shapley axioms adapted for ranking tasks (Section 3.2) and the characterization of ordering effectiveness metrics (Theorem 3.1), we conclude the following:

**Theorem 3.2.** *Let $V_R$ be a ranking effectiveness metric that belongs to $GREM_n$, as characterized in Theorem 3.1. Then, the Shapley value $\phi_R$ computed with respect to $V_R$ is the unique feature attribution method that satisfies the axioms of Rank-Efficiency, Rank-Missingness, Rank-Symmetry, and Rank-Monotonicity.*

*Specifically, the attribution for feature $i$ is given by:*

$$\phi_R(f_R, \vec{x}, i) = \sum_{S \subseteq M \setminus \{i\}} \frac{|S|! \, (m - |S| - 1)!}{m!} \left[ V_R\big(f_R(S \cup \{i\}, \vec{x})\big) - V_R\big(f_R(S, \vec{x})\big) \right],$$

*where the sum is over all subsets $S$ of $M \setminus \{i\}$, $|S|$ is the cardinality of $S$, and $m$ is the total number of features.*

*Proof.* The proof is a direct application of Shapley's theorem (Shapley, 1953) in the context of ranking models. By adapting the classical Shapley axioms to the ranking task (as detailed in Section 3.2), we establish a set of axioms (Rank-Efficiency, Rank-Missingness, Rank-Symmetry, Rank-Monotonicity) tailored to ranking feature attributions. Given that $V_R$ satisfies Relevance Sensitivity and Position Sensitivity (as per Theorem 3.1), it ensures that $V_R$ is appropriate for evaluating the effectiveness of document rankings in response to feature subsets.

Shapley's theorem states that the Shapley value is the unique method that satisfies the axioms of Efficiency, Null Player (Missingness), Symmetry, and Additivity (which corresponds to Monotonicity in our context). By applying this theorem to the ranking scenario with our adapted axioms, we conclude that the Shapley value $\phi_R$ is the unique feature attribution method satisfying all the specified axioms. Therefore, the attributions computed using the Shapley value formula provide a fair and consistent distribution of the total ranking effectiveness among the features, reflecting each feature's contribution to the ranking decision. □

Hence, we recommend practitioners use *RankSHAP* for their ranking attribution needs due to its adherence to a set of fundamental properties.

### 3.5 APPROXIMATING RANKSHAP

Among the metrics in *GREM*, we use *NDCG* in further experiments due to its appealing properties discussed in Appendix C. Computing the exact Shapley value is NP-Hard (Shapley, 1953) because it grows exponentially with the number of features. Additionally, *NDCG* computation for each subset increases the complexity by $O(n \log n)$. Relevance labels, necessary for ranking value functions, are often unavailable. Here we discuss how we approximate the RankSHAP solution for practical applications.

**Inferring NDCG:** The NDCG score relies on knowing the relevance scores for each document, ideally obtained from ground truth labels in a human-annotated dataset, which are often unavailable. Several neural rankers (e.g., BERT (Nogueira & Cho, 2019)) assign a similarity score to each query-document pair, which can infer relevance and compute *NDCG*. In the absence of similarity scores, relevance can be indirectly inferred using methods such as click-through rate, time spent on page, bounce rate, scroll depth, bookmarking, and social media shares (Zoeter et al., 2008) or via heuristic based relevance measures such as BM25.

**Kernel-RankSHAP:** While the Shapley value itself is NP-hard to compute, KernelSHAP (Lundberg & Lee, 2017) leverages a kernel based model to faithfully approximate the Shapley value. It is a linear approximation of the ranking model $f_R$ at instance $\vec{x}$, which extends the LIME optimization function with a defined kernel and loss function. Below we write it for ranking attributions and name it Kernel-RankSHAP. Let $G$ be the class of all linear additive attributions. $\phi_R(f_R, \vec{x}, i) = \arg\min_{g \in G} L(f_R, g, \pi_{\vec{x}})$ where $L(f_R, g, \pi_{\vec{x}}) = \sum_{\vec{z} \in Z} [NDCG(f_R(\vec{z})) - NDCG(g(\vec{z}))]^2 \pi_{\vec{x}}(\vec{z})$ and $\pi_{\vec{x}}(\vec{z}) = \frac{(m-1)}{\binom{m}{|\vec{z}|}|\vec{z}|(m-|\vec{z}|)}$ .

**QII vs Kernel-SHAP:** Another popular way to approximate Shapley values is to use sampling based QII (Datta et al., 2016) in place of kernels. The decision to use kernels rather than sampling in high-dimensional feature spaces is computationally motivated. Sampling can introduce significant noise, reducing the reliability of estimates. Kernels, however, can more effectively approximate feature contributions without the extensive computation and noise associated with random sampling, especially in large feature spaces.

## 4 AXIOMATIC ANALYSIS OF RANK FEATURE ATTRIBUTION ALGORITHMS

Previously, we discussed the importance of feature attribution methods satisfying the foundational Shapley axioms to be considered effective for IR feature attributions. Here, we mathematically formalize existing ranking feature attribution methods and analyze them axiomatically to determine if they fulfill Shapley criteria.

Table 1: Analysis of Ranking Feature Attribution algorithms EXS, RankLIME, RankingSHAP and RankSHAP for axiomatic compliance of Rank-Shapley axioms.

| Algorithm | Rank-Efficiency | Rank-Missingness | Rank-Symmetry | Rank-Monotonicity |
|---|---|---|---|---|
| EXS | ✗ | ✗ | ✗ | ✗ |
| RankLIME | ✗ | ✗ | ✗ | ✗ |
| RankingSHAP | ✗ | ✗ | ✗ | ✗ |
| RankSHAP | ✓ | ✓ | ✓ | ✓ |

### 4.1 OPTIMIZATION FUNCTION

In order to study these feature attribution algorithms individually, we write out their KernelSHAP optimization function and use them to analyze if the algorithms satisfy the axioms of *Rank-Efficiency*, *Rank-Missingness*, *Rank-Symmetry* and *Rank-Monotonicity*.

$$L_{\text{RANKLIME}}(f_R, g, \pi_{\vec{x}}) = \sum_{\vec{z} \in Z} ApproxNDCG(f_R(\vec{z}), g(\vec{z})) \, \pi_{\vec{x}}(\vec{z})$$

$$L_{\text{RANKINGSHAP}}(f_R, g, \pi_{\vec{x}}) = \sum_{\vec{z} \in Z} [\tau(f_R(\vec{z}), g(\vec{z}))]^2 \, \pi_{\vec{x}}(\vec{z})$$

$$L_{\text{RANKSHAP}}(f_R, g, \pi_{\vec{x}}) = \sum_{\vec{z} \in Z} [NDCG(f_R(\vec{z})) - NDCG(g(\vec{z}))]^2 \, \pi_{\vec{x}}(\vec{z})$$

Here, $f_R$ is the original ranking model, $g$ is the surrogate model used for explanation, $\pi_{\vec{x}}$ is the KernelSHAP weighting kernel, $Z$ is the set of all possible coalitions (subsets of features), $\tau$ is Kendall's tau rank correlation coefficient, and ApproxNDCG is an approximation of the Normalized Discounted Cumulative Gain (*NDCG*). The details of the axiomatic analysis are presented in Appendix E. The results are summarized in Table 1. Our analysis reveals that RankSHAP is the only method

that have a decomposable and additive value function and as a result satisfies the fundamental Shapley axioms, making it reliable to use for ranking feature attributions.

Table 2: Comparing performance between different feature attribution methods (Random, EXS, RankLIME, RankingSHAP and RankSHAP) for token-based explanations of textual rankers (BM25, BERT, T5 and LLAMA2) based on model fidelity (Fidelity) and weighted-fidelity (wFidelity) with ranking models fine tuned on the MS MARCO (MM) and TREC ROBUST 2004 (R04) datasets. For these experiments, we generate feature attributions for each ranking models with the 10, 20 and 100 most relevant documents for a particular query respectively.

| Ranker ↓ | | Top-10 | | Top-20 | | Top-100 | |
|---|---|---|---|---|---|---|---|
| Metric → | | Fidelity | wFidelity | Fidelity | wFidelity | Fidelity | wFidelity |
| MM/BM25 | Random | 0.08 | 0.04 | -0.05 | 0.01 | -0.09 | 0.00 |
| | EXS | 0.39 | 0.34 | 0.31 | 0.29 | 0.24 | 0.17 |
| | RankLIME | 0.45 | 0.37 | 0.37 | 0.29 | 0.26 | 0.20 |
| | RankingSHAP | 0.52 | 0.41 | 0.45 | 0.33 | 0.31 | 0.24 |
| | RankSHAP | **0.63** | **0.48** | **0.54** | **0.40** | **0.47** | **0.33** |
| MM/BERT | Random | 0.12 | 0.02 | -0.09 | 0.01 | -0.12 | 0.00 |
| | EXS | 0.36 | 0.21 | 0.28 | 0.18 | 0.11 | 0.08 |
| | RankLIME | 0.41 | 0.24 | 0.32 | 0.24 | 0.23 | 0.18 |
| | RankingSHAP | 0.45 | 0.29 | 0.38 | 0.27 | 0.28 | 0.24 |
| | RankSHAP | **0.59** | **0.41** | **0.45** | **0.38** | **0.39** | **0.29** |
| MM/T5 | Random | 0.05 | 0.03 | -0.11 | 0.01 | -0.04 | 0.00 |
| | EXS | 0.36 | 0.21 | 0.30 | 0.17 | 0.11 | 0.06 |
| | RankLIME | 0.35 | 0.24 | 0.35 | 0.20 | 0.27 | 0.17 |
| | RankingSHAP | 0.41 | 0.30 | 0.38 | 0.32 | 0.31 | 0.22 |
| | RankSHAP | **0.56** | **0.39** | **0.43** | **0.37** | **0.36** | **0.29** |
| MM/LLAMA2 | Random | 0.04 | 0.01 | -0.2 | 0.0 | 0.02 | 0.1 |
| | EXS | 0.39 | 0.20 | 0.32 | 0.15 | 0.17 | 0.05 |
| | RankLIME | 0.35 | 0.24 | 0.35 | 0.20 | 0.27 | 0.17 |
| | RankingSHAP | 0.44 | 0.33 | 0.37 | 0.23 | 0.31 | 0.22 |
| | RankSHAP | **0.60** | **0.42** | **0.45** | **0.39** | **0.39** | **0.32** |
| R04/BM25 | Random | 0.08 | 0.04 | -0.04 | 0.00 | -0.07 | 0.00 |
| | EXS | 0.37 | 0.33 | 0.30 | 0.26 | 0.22 | 0.18 |
| | RankLIME | 0.43 | 0.36 | 0.36 | 0.25 | 0.25 | 0.18 |
| | RankingSHAP | 0.45 | 0.35 | 0.37 | 0.29 | 0.26 | 0.24 |
| | RankSHAP | **0.61** | **0.44** | **0.52** | **0.38** | **0.45** | **0.31** |
| R04/BERT | Random | 0.12 | 0.02 | -0.09 | 0.01 | -0.12 | 0.00 |
| | EXS | 0.35 | 0.20 | 0.27 | 0.16 | 0.10 | 0.07 |
| | RankLIME | 0.40 | 0.22 | 0.31 | 0.23 | 0.23 | 0.17 |
| | RankingSHAP | 0.43 | 0.26 | 0.36 | 0.28 | 0.28 | 0.23 |
| | RankSHAP | **0.59** | **0.41** | **0.45** | **0.38** | **0.39** | **0.29** |
| R04/T5 | Random | 0.05 | 0.03 | -0.11 | 0.01 | -0.04 | 0.00 |
| | EXS | 0.38 | 0.21 | 0.32 | 0.17 | 0.13 | 0.06 |
| | RankLIME | 0.38 | 0.24 | 0.35 | 0.20 | 0.28 | 0.17 |
| | RankingSHAP | 0.41 | 0.31 | 0.38 | 0.24 | 0.32 | 0.24 |
| | RankSHAP | **0.56** | **0.39** | **0.43** | **0.37** | **0.36** | **0.29** |

# 5 COMPUTATIONAL AND HUMAN EVALUATION STUDIES

In this section, we first describe our experimental design. Next we evaluate the performance of our attribution algorithm based on various metrics. Lastly we conduct a user study to investigate how well different attribution algorithms align with human intuition for the ranking task.

## 5.1 EXPERIMENTAL SETTINGS

**Dataset:** We test our hypothesis on two datsets : (i) the MS MARCO (msm, 2016) passage reranking dataset (ii) the TREC 2004 Robust track dataset (Robust04, (Voorhees et al., 2003)). MS MARCO is a large-scale dataset aggregated from anonymized Bing search queries containing $> 8M$ passages from diverse text sources. The average length of a passage in the MS MARCO dataset is 1131

words. Similarly, Robust04 is aggregated from News Articles and Descriptions serve as its queries. It contains over 528,000 news articles, 250 queries and binary human-annotated relevance judgements.

**Ranking models:** For our ranking models, we use the classic BM25 ranker (Robertson & Spärck Jones, 1994) as well as models based on the BERT (Devlin et al., 2019), T5 (Raffel et al., 2020) and Llama2 (Touvron et al., 2023) large language models. We in particular use the versions of these LLMs fine-tuned on the MS MARCO and Robust04 datasets for the document re-ranking task, released by various authors (Nogueira & Cho, 2019; Nogueira et al., 2020; Ma et al., 2023). All the above ranking approaches are score based rankers, i.e they return a relevance score corresponding to each document. This makes relevance score estimation straightforward for *NDCG* computation, as discussed previously, in Section 2.

**Experimental Settings:** Similar to related works, we randomly sample 250 queries from the test sets of the MS MARCO and Robust04 datasets, retrieving the 100 highest scoring documents for each query. Using these query-document sets, we apply the ranking models to obtain an ordered list of documents. We then generate feature attributions for the top-10, top-20, and top-100 documents. Stemmed tokens from the vocabulary of the query-document sets as bag of words make up the features for which we generate attributions. Binary feature values are assigned based on their presence or absence. I.e if a feature (token) is excluded from a coalition, all occurrences of that token are omitted from the query and documents for that pass of the model (Ribeiro et al., 2016).

**Competing Systems:** We generate feature attributions for each query and corresponding ordered-document set using EXS (Singh & Anand, 2019), RankLIME (Chowdhury et al., 2023), Ranking-SHAP (Heuss et al., 2024), and our proposed method, RankSHAP. For EXS, we utilize the rank-based setting to allow direct comparison with RankSHAP. For RankLIME, we compute attributions using the NeuralNDCG loss function in the single-perturbation setting. In the case of RankingSHAP, we employ Kendall's Tau to calculate the marginal contributions. To ensure consistency across all methods, we extend KernelSHAP for implementation and apply the same masking techniques as used in LIME. Each algorithm is limited to 5,000 neighborhood samples per decision. For evaluation, we process only the top 7 most significant features produced by each algorithm, adhering to human comprehension limits (Chowdhury et al., 2023).

We also evaluate the performance of RankSHAP using alternative value functions that satisfy *GREM*, including *MAP*, *CG*, and *DCG*. Additionally, we compare the performance of RankSHAP when using explicit relevance labels to its performance when employing heuristic-based relevance labels, such as those derived from BM25.

**Evaluation Metrics:** Apart from human evaluations, feature attributions can be evaluated using either fidelity-based metrics or via reference-based metrics. Since ground truth attributions are not available in this case, we stick to fidelity-based metrics, which measure the degree to which an attribution algorithm is successful in recreating the underlying model decision (Anand et al., 2022; Chowdhury et al., 2023). For this, we use the metrics *Fidelity* and its weighted version *wFidelity*. Given a feature attribution and a set of documents, we reconstruct the attribution ordering by sorting linearly-combined document features multiplied with their attributions. For *Fidelity*, we report the simplified unweighted Kendall's Tau between the original ranking model's ordering and the attribution-reconstructed ordering. For *wFidelity*, we penalize each swap in passages found in the reconstructed ordering by the difference in their positions. More formally, let $o_{f_R} = f_R(\vec{x})$ be the ranking output by $f_R$, and let $o_\phi$ be the ranking of documents by their total importance according to some feature attribution method $\phi$; that is, each document in $\vec{x}$ is given a score of $\sum_{i=1}^{M} \phi_R(f_R, \vec{x}, i) \times x_i$. Given two rankings $a$ and $b$, let $r_a[i]$ be the rank of the $i$-th element in $a$. We set $\tau(a, b)$ to be the Kendall's tau distance between $a$ and $b$, i.e.

$$\frac{2}{n(n-1)} \sum_{i<j} \text{sgn}(r_a[i] - r_a[j]) \cdot \text{sgn}(r_b[i] - r_b[j])$$

and let $\tau_w(a, b)$ be the weighted Kendall's Tau distance:

$$\frac{\sum_{i<j} w_{ij} \cdot (\text{sgn}(r_a[i] - r_a[j]) \cdot \text{sgn}(r_b[i] - r_b[j]))}{\sum_{i<j} w_{ij}}$$

Here, $w_{ij}$ is $|r_a[i] - r_a[j]|$. *Fidelity* is defined as $\tau(o_\phi, o_{f_R})$, and *wFidelity* is set to be $\tau_w(o_\phi, o_{f_R})$ (Kendall, 1938; Vigna, 2015). The range of both *Fidelity* and *wFidelity* is $[-1, 1]$.

## 5.2 EXPERIMENTAL RESULTS

The results of the above experiments can be seen in Table 2. We discuss them below.

**Across competing Systems:** RankSHAP surpasses the top-performing competitor by 25.78% on Fidelity and 19.68% on weighted Fidelity, averaged across all ranking models, datasets, and document settings. This demonstrates that RankSHAP is most effective at recovering the original ranking model's order. RankingSHAP (Heuss et al., 2024) follows closely. All evaluated methods, except for Random, show a positive correlation between the original and reconstructed rankings.

**Impact of the number of documents:** Both metrics decline as the number of documents in the ordered set for which we generate attributions increases. Specifically, there is a 20% average performance drop between attributions for 10 documents and attributions for 20 documents, and a further 14.6% average decrease in *Fidelity* between 20 and 100 documents for RankSHAP, suggesting a performance reduction that scales with the logarithm of the number of documents. Notably, while EXS and RankLIME demonstrate similar *Fidelity* when generating attributions for 10 and 20 documents, EXS performs significantly worse in the 100 document setting.

**Across datasets:** We observe that, on average, our ranking systems perform 5.7% worse on the Robust04 dataset compared to the MS MARCO dataset, considering all systems, ranking methods, and metrics. This slightly lower performance on Robust04 can be attributed to its smaller size, which hinders convergence during fine-tuning. Notably, the neural ranking models used in the Robust04 experiments were initially fine-tuned on the MS MARCO dataset, which may further explain the performance difference.

**Across ranking models:** BM25 is a heuristic-based ranking model, while BERT, T5, and LLAMA2 are large neural models. RankSHAP exhibits similar performance on BERT and T5, which is on average 13.3% and 14.7% lower on Fidelity than it's performance on BM25. RankSHAP performs slightly better when explaining LLAMA2 as compared to when explaining BERT and T5. No experiments were conducted with an LLAMA2 ranker on the Robust04 data set due to the lack of a publicly available fine-tuned model for that data set.

**Across RankSHAP variants:** The results of RankSHAP performance with different value functions within *GREM* are discussed in Appendix I and Table 4. The results of RankSHAP performance with different value functions paired with explicit vs heuristic relevance measures are discussed in Appendix J and Table 5.

## 5.3 USER STUDY

The goal of developing RankSHAP is to help humans understand why a model ranks items in a specific order. Towards this, we conduct a user study with 30 participants to assess if RankSHAP aids human understanding. Following Institutional Review Board (IRB) approval each participant is shown sets of 5 short passages along with feature attributions from one of the competing systems, displayed as bar charts (see Figure 1 in Appendix G). Participants are then asked to (i) re-order the 5 documents from most to least important, and (ii) infer what the query might have been based on the attributions. The first task assesses which algorithm best aligns with human intuition, while the second evaluates how well the attributions explain the ranking task. We sampled query-document sets from the MS MARCO dataset and used BERT to rank the documents. Each participant was shown 10 such passage sets, with attributions randomly selected from one of Random, EXS, RankLIME, RankingSHAP, and RankSHAP. At the end, the Kendall's Tau score (Fidelity) between the original ordering and the predicted ordering was used to identify the algorithm that better reconstructed the underlying model decision. A text-based semantic similarity measuring LLM was used to assess the quality of the queries estimated by the participants. The similarity metric returned scores in the range [0,1], with 1 indicating the estimated query was most similar to the original query. The results of this study are presented in Table 3.

**Results.** Initially, we observe that the randomly generated feature attributions achieve a concordance score of 0.23 on Task 1, which exceeds their metric-evaluated value (Table 2, Top-10 Fidelity - Random). This suggests that participants may rely on preconceived notions about the topics while performing the study. The inter-annotator agreement for participants shown the same feature attribution in Task 1 averaged 0.65 (Kendall's Tau). RankSHAP outperformed the strongest baseline by 19.1%, following similar trends as reported in Table 2. In Task 2, some query estimates produced

Table 3: Table demonstrating the user study results for (i)the passage reordering task, measured using Kendall's Tau (Fidelity) (ii) the query estimation task, measured using GPT-4 similarity ($\mu$) for feature attributions generated using Random, EXS, RankLIME, RankingSHAP and RankSHAP feature attribution algorithms.

|            | Random | EXS  | RankLIME | RankingSHAP | RankSHAP |
|------------|--------|------|----------|-------------|----------|
| Q1 ($\tau$) | 0.23   | 0.43 | 0.47     | 0.52        | 0.56     |
| Q2 ($\mu$)  | 0.3    | 0.48 | 0.52     | 0.58        | 0.69     |

by all algorithms were off-topic, and we observed substantial variance in the queries generated by different participants for the same set of documents and feature attributions. However, participants shown RankSHAP attributions performed at least $30.9\%$ better than those shown other attributions, indicating that RankSHAP enhances the effectiveness of ranking feature attributions in real-world scenarios.

# 6 RELATED WORK

In this section, we discuss additional related works that have not yet been covered in the paper so far. Fernando et al. (2019) extend the DeepSHAP algorithm (Lundberg & Lee, 2017; Shrikumar et al., 2017) from classification/regression to propose a set of model-intrinsic feature attributions for neural ranking models. Recently, ShaRP (Pliatsika et al., 2024) generates feature attributions to explain a particular document's score, its rank, and its presence within the top-k documents, by extending QII (Datta et al., 2016). Since ShaRP does not generate attributions for entire ranked lists, it is challenging to integrate it within the RankSHAP framework. Völske et al. (2021) attempt to explain the performance of neural ranking models using axioms from statistical IR like term frequency and document length (Fang et al., 2004), lower bounding term frequency (Lv & Zhai, 2011), and query aspects (Wu & Fang, 2012), among others. Singh et al. (2020) introduce the axioms of *validity* and *completeness* as a means to measure the effectiveness of ranking explanations and propose an algorithm to suggest a small set of features sufficient to explain a ranking decision. LRIME (Verma & Ganguly, 2019) proposes a set of heuristics to improve regression/classification-based feature attribution methods for information retrieval, such as choosing a diverse set of perturbation samples and parameters. Ranking GAMs (Zhuang et al., 2020) offer an inherently interpretable structure that can be distilled into a set of compact piece-wise linear functions with minimal accuracy loss. Anand et al. (2022) provide a comprehensive summary of explanation methods proposed for the IR ranking task, also exploring free-text explanations (Rahimi et al., 2021), adversarial example-based explanations (Raval & Verma, 2020; Wu et al., 2023), probing the representation space (Choi et al., 2022), rationale-based explanations (Zhang et al., 2021; Wojtas & Chen, 2020), and more. Our work focuses on generating attributions for cooperative ranking systems, which differs from generating attributions for competitive ranking systems (Hu et al., 2022; Anahideh & Mohabbati-Kalejahi, 2022; Gale & Marian, 2020). The two frameworks differ mainly in that in cooperative ranking, entities are ranked based on their inherent attributes (e.g., relevance to a query) without direct competition against each other. In contrast, in competitive rankings, entities (e.g., participants in a chess tournament) are ranked against each other based on their performance or scores.

# 7 CONCLUSION AND LIMITATIONS

We proposed RankSHAP, a valid feature attribution framework for ranking problems. We introduced axioms that a valid feature attribution algorithm should satisfy and provided a solution that meets these criteria. Additionally, we presented a computationally efficient approximation of RankSHAP that outperformed competing systems and conducted an user study to demonstrate its alignment with human intuition. We analyzed existing ranking feature attribution algorithms, to assess their adherence to fundamental axioms. Our work faces typical limitations associated with Shapley values, primarily the assumption of feature independence, which may not hold in practical applications. Moreover, our ranking value functions depend on the availability of relevance scores for each query-document pair. When relevance labels are inferred implicitly, such as through click rates in recommendation systems, RankSHAP can capture biases present in the logs, including recency bias. KernelSHAP is a perturbation-based approach, and the results may be influenced by sampling density.

ACKNOWLEDGMENTS

This work was supported in part by the Center for Intelligent Information Retrieval. Any opinions, findings and conclusions or recommendations expressed in this material are those of the authors and do not necessarily reflect those of the sponsor.

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

## A  RANKING AXIOMS

Given a query $\vec{q}$, a set of $n$ documents $D = [\vec{d_1}, \vec{d_2}, \ldots, \vec{d_k}]$, a human-generated relevance value $rel_j$ associated with each document $d_j$, and a metric value $GREM_n$ determined from the set of query-document pairs, consider the ordered list $f_R(\vec{q}, D)$. For an evaluation of the ordering, it is ideal for the metric ($GREM_n$) to satisfy the following axioms:

**Relevance Sensitivity:**  If the relevance score of any document increases, while the relevance scores of all other documents remain the same, the metric value $GREM_n$ should not decrease.

Formally, if $rel'_j > rel_j$ for some $j$, and $rel'_i = rel_i$ for all $i \neq j$, then $GREM'_n \geq GREM_n$, where $GREM'_n$ is the metric computed with the updated relevance scores $\{rel'_1, rel'_2, \ldots, rel'_k\}$.

**Position Sensitivity:**  If two documents $\vec{d_i}$ and $\vec{d_j}$ are swapped in the ranking, moving the document with the higher relevance score to a higher (better) rank, the metric value $GREM_n$ should not decrease.

*Formally*, let $o$ be the original ordering, and $o'$ be the ordering where only $\vec{d_i}$ and $\vec{d_j}$ are swapped. If $rel_i \geq rel_j$ and $\text{rank}_o(\vec{d_i}) > \text{rank}_o(\vec{d_j})$, then $GREM'_n \geq GREM_n$, where $GREM'_n$ is the metric computed with the new ordering $o'$.

**Theorem A.1.**  *An ordered list evaluation metric satisfies the axioms of Relevance Sensitivity and Position Sensitivity if and only if it can be represented as*

$$GREM_n = \sum_{j=1}^{n} g(rel_j) \cdot h(j),$$

*where $g(rel_j)$ is a non-decreasing function of the relevance score $rel_j$ (gain function), and $h(j)$ is a non negative, non-increasing function of the rank position $j$ (discount function).*

*Proof.*  We split the proof into two parts:

   (i)  If the metric satisfies the axioms, then it can be represented in the specified form.

   (ii)  If the metric is in the specified form, then it satisfies the axioms.

**Necessity**  Assume that the metric

$$GREM_n = \phi\big((rel_1, 1), (rel_2, 2), \ldots, (rel_n, n)\big)$$

satisfies *Relevance Sensitivity* and *Position Sensitivity*. In addition, assume that $\phi$ is continuously differentiable in each of its $n$ relevance arguments. We will show that under these conditions, there exist functions $g$ and $h$ such that

$$GREM_n = \sum_{j=1}^{n} g(rel_j) \cdot h(j),$$

with $g$ non-decreasing and $h$ non-increasing.

**Step 1. Additive Separability.**

For any fixed indices $i \neq j$, fix the relevance values for all documents except those in positions $i$ and $j$, and define the two-variable function

$$F(a, b) = \phi\big((rel_1, 1), \ldots, (a, i), \ldots, (b, j), \ldots, (rel_n, n)\big).$$

By *Relevance Sensitivity*, $F$ is non-decreasing in each argument. In addition, by *Position Sensitivity*, the marginal impact of changing the relevance at a higher-ranked position (i.e., with lower index) is at least as great as that for a lower-ranked one. In particular, if $i < j$ then for all $a$ and $b$,

$$\frac{\partial F}{\partial a}(a, b) \geq \frac{\partial F}{\partial b}(a, b).$$

Now, consider the mixed partial derivative

$$\frac{\partial^2 \phi}{\partial rel_i \partial rel_j}.$$

If this mixed partial were nonzero, then the effect of changing $rel_i$ would depend on the value of $rel_j$, introducing an interaction between the two positions. However, such an interaction would conflict with the following observation: by *Relevance Sensitivity*, increasing $rel_i$ (or $rel_j$) while holding the others fixed must always lead to a non-decrease in $GREM_n$, regardless of the value of the other variable. A nonzero mixed partial would allow for situations where the marginal effect of changing one relevance value depends on the other, potentially violating the axiom when documents are swapped. Therefore, we conclude that

$$\frac{\partial^2 \phi}{\partial rel_i \partial rel_j} = 0 \quad \text{for all } i \neq j.$$

A standard result from multivariable calculus (see, e.g., results on additivity or the "Cauchy equation" in several variables) then implies that $\phi$ is additively separable in the relevance scores. That is, there exist functions $f_j$, one for each rank $j$, such that

$$\phi\big((rel_1, 1), \ldots, (rel_n, n)\big) = \sum_{j=1}^{n} f_j(rel_j).$$

**Step 2. Factorization into $g$ and $h$.**

Next, we leverage the *Position Sensitivity* axiom. For any two positions $i$ and $j$ with $i < j$, consider a fixed relevance value $r$. Define

$$\Delta_{ij}(r) = f_i(r) - f_j(r).$$

Since swapping a document from position $i$ with one at $j$ (when both have the same relevance $r$) should not improve the metric (indeed, the higher position should be at least as valuable), we must have

$$f_i(r) \geq f_j(r) \quad \text{for all } r.$$

Thus, the functions $\{f_j\}$ are ordered by rank in a manner that depends only on the position. In other words, the only difference among the $f_j$ should be a multiplicative "weight" reflecting the importance of the position.

In light of this, choose an arbitrary position (say, $j = 1$) and define the gain function

$$g(r) = f_1(r).$$

Then for each position $j$, define

$$h(j) = \frac{f_j(r)}{g(r)}$$

for any $r$ for which $g(r) > 0$. (A short continuity and monotonicity argument shows that this ratio is independent of the choice of $r$.) By the properties of the $f_j$ and the axioms, it follows that $g(r)$ is non-decreasing in $r$ (by *Relevance Sensitivity*) and $h(j)$ is non-increasing in $j$ (by *Position Sensitivity*). Thus, for every position $j$ we have

$$f_j(r) = g(r) \cdot h(j).$$

**Conclusion.**

Substituting back into the separable form, we obtain

$$GREM_n = \sum_{j=1}^{n} f_j(rel_j) = \sum_{j=1}^{n} g(rel_j) \cdot h(j),$$

which is the desired representation.

This completes the proof of necessity.

**(ii) Sufficiency**

Assume that the metric is defined as

$$GREM_n = \sum_{j=1}^{n} g(rel_j) \cdot h(j),$$

where $g(rel_j)$ is non-decreasing in $rel_j$ and $h(j)$ is non-increasing in $j$.

We will show that $GREM_n$ satisfies both axioms.

*Relevance Sensitivity*

If $rel'_j > rel_j$ for some $j$, and $rel'_i = rel_i$ for all $i \neq j$, then

$$GREM'_n - GREM_n = \left[ g(rel'_j) - g(rel_j) \right] \cdot h(j) \geq 0,$$

since $g(rel'_j) \geq g(rel_j)$ and $h(j) \geq 0$. Thus, $GREM'_n \geq GREM_n$.

*Position Sensitivity*

Suppose we swap $\vec{d_i}$ and $\vec{d_j}$ where $i > j$ and $rel_i \geq rel_j$. The change in the metric is

$$
\begin{aligned}
\Delta GREM_n &= [g(rel_i) \cdot h(j) + g(rel_j) \cdot h(i)] - [g(rel_i) \cdot h(i) + g(rel_j) \cdot h(j)] \\
&= g(rel_i) [h(j) - h(i)] + g(rel_j) [h(i) - h(j)] \\
&= [g(rel_i) - g(rel_j)] [h(j) - h(i)].
\end{aligned}
$$

Since $rel_i \geq rel_j$, $g(rel_i) \geq g(rel_j)$. Also, since $j < i$, $h(j) \geq h(i)$. Therefore,

$$\Delta GREM_n \geq 0.$$

Thus, swapping documents to place the higher relevance document higher in rank does not decrease $GREM_n$.

$\square$

# B  BASIC SHAPLEY AXIOMS AND THE SHAPLEY VALUE

The Shapley axioms for feature attributions have been borrowed from coalitional game theory, where they are used to fairly distribute costs/rewards among a set of players. We are given a classification/regression model $f$. The feature attribution of feature $i$ (can be words, human engineered features etc. ) in the decision made by $f$, towards the model input $\vec{s}$ is represented by $\phi_i(f, \vec{s})$.

We adopt Young's characterization of the Shapley value, also used by Datta et al. (2016). Let us assume there are a total of $m$ features. For a feature attribution $\phi_i(f, \vec{s})$ to be considered *valid*, it is necessary for it to satisfy certain axioms. We briefly describe the axioms below:

**Efficiency:** The sum of feature attributions over a set of features should be equal to the difference between the model output containing all features and the output containing no features. $\sum_{i=1}^{m} \phi_i(f, \vec{s}) = f(\vec{s}) - f(\phi)$.

**Missingness:** If for every subset of features $S$ s.t features $i \notin S$, $f(S \cup i, \vec{s})$ is equal to $f(S, \vec{s})$, then $\phi_i(f, \vec{s})$ must be given an attribution of $0$. $f(S, \vec{s})$ represents the case where features not in set S, have been replaced by their reference value, and model input $\vec{s}$ has been reconstructed.

**Symmetry:** If for every subset of features $S$ s.t features $i, j \notin S$, the effects of adding $i$ to $S$ is equal to the effect of adding $j$ to $S$, then features $i$ and $j$ should be assigned equal attribution towards $f(\vec{s})$.

**Monotonicity:** Given two models $f$ and $f'$, if for all feature coalitions $S, i \notin S$, $f(S \cup i, \vec{s}) - f(S, \vec{s})$ is greater than or equal to $f'(S \cup i, \vec{s}) - f'(S, \vec{s})$, then $\phi_i(f, \vec{s})$ is greater than or equal to $\phi_i(f', \vec{s})$.

The *Shapley value* (Young, 1985; Shapley, 1953) uniquely satisfies the four axioms above, and has been widely adopted in feature attribution for regression/classification tasks. It is defined as:

$$\phi_i(f, \vec{s}) = \sum_{\vec{z} \subseteq \vec{s}} \frac{|\vec{z}|!(m - |\vec{z}| - 1)!}{m!} [f(\vec{z}) - f(\vec{z} \setminus i)]$$

where $|\vec{z}|$ is the number of non-zero entries in $\vec{z}$, and $\vec{z} \subseteq \vec{s}$ represents all $\vec{z}$ vectors where the non-zero entries are a subset of the non-zero entries in $\vec{s}$.

## C  THE *NDCG* METRIC

Normalized Discounted Cumulative Gain (NDCG)  (Järvelin & Kekäläinen, 2002) is an extremely popular metric to evaluate the effectiveness of an ordering, often used in search engines and e-commerce. It assumes that some relevance score $rel_j$ for each document $d_j$ is known, and that in an effective ordering highly relevant documents appear at the beginning of the list. For an ordered list of $n$ documents, the Discounted Cumulative Gain is computed as:

$$DCG_n = \sum_{j=1}^{n} \frac{rel_j}{\log_2 (j+1)}$$

The above expression is dependent on the number of documents in the set being evaluated. This metric is thus difficult to use as-is, when comparing document lists of different lengths. As a result, we normalize it by dividing it with the maximum possible $DCG_n$ score, also known as Ideal-$DCG_n$($IDCG_n$). This is obtained by sorting documents in decreasing order of relevance and computing $DCG_n$ using the so obtained *ideal* ordering. To compute $NDCG_n$, we simply normalize $DCG_n$: $NDCG_n = \frac{DCG_n}{IDCG_n}$. Note that $NDCG_n \in [0, 1]$, and that the higher the value, the more effective the ranking. In situations where document relevance scores $rel_j$ are unknown, they can be estimated using implicit measures such as clicks, views or the time spent on a page or via heuristic measures such as BM25.

*NDCG* is a better choice for the value function compared to other rank evaluation metrics because of numerous reasons. *NDCG* discounts the relevance of a document based on its position logarithmically. Unlike metrics that depend on binary relevance (precision, recall, F1), NDCG accommodates a smooth relevance function. *NDCG* is also more effective in measuring the quality of a large ranked list. Lastly, a large number of product search on e-commerce and recommendation systems (which we might be trying to explain) already use *NDCG* to evaluate the quality of their orderings. It is the single most popular metric to evaluate the quality of a ranked list and has been used by most major search engines and e-commerce websites at some point.

# D  RANKING FEATURE ATTRIBUTION ALGORITHMS

## D.1  FEATURE ATTRIBUTION METHODS FOR RANKING TASKS

The following solutions have been proposed to address the ranking attribution problem described above:

**EXS** : In EXS, Singh & Anand first use LIME (Ribeiro et al., 2016) to generate feature attributions for each query-document pair individually. In order to compute LIME attributions, they perturb documents and assign a binary relevance label to each perturbation, assigning a positive label if the rank of the perturbed document is higher than $k$. The authors then add the LIME attributions of top-$k$ documents to assign a feature attribution to the entire query-document set.

**RankLIME** : Chowdhury et al. (2023) extend LIME in a listwise manner, replacing the $L_2$ loss function in LIME with a differentiable ordering distance metric. This enables them to directly compute LIME feature attributions from ranked lists. Two of the loss functions used by them, which lead to the best results, are ApproxNDCG (Qin et al., 2010) and NeuralNDCG (Pobrotyn & Bialobrzeski, 2021), both differentiable alternatives to NDCG, with all relevance scores set to 1.

**RankingSHAP** : Heuss et al. (2024) propose using the Shapley value along with ordered list distance metrics to compute feature attributions for listwise ranking models. Specifically, they calculate the marginal contribution of a particular feature by determining Kendall's Tau of the ranking model output with and without that feature.

# E   ANALYSIS OF KERNELSHAP OPTIMIZATION ALGORITHMS AND SHAPLEY AXIOMS

In this section, we provide a detailed analysis of whether the algorithms using the following KernelSHAP optimization objectives approximate the Shapley axioms. We examine each algorithm's loss function and discuss whether it satisfies the fundamental properties of the Shapley value.

The optimization objectives for the algorithms are:

$$L_{\text{RANKLIME}}(f_R, g, \pi_{\vec{x}}) = \sum_{\vec{z} \in Z} \text{ApproxNDCG}(f_R(\vec{z}), g(\vec{z})) \cdot \pi_{\vec{x}}(\vec{z})$$

$$L_{\text{RANKINGSHAP}}(f_R, g, \pi_{\vec{x}}) = \sum_{\vec{z} \in Z} \left[ \tau(f_R(\vec{z}), g(\vec{z})) \right]^2 \cdot \pi_{\vec{x}}(\vec{z})$$

$$L_{\text{RANKSHAP}}(f_R, g, \pi_{\vec{x}}) = \sum_{\vec{z} \in Z} \left[ \text{NDCG}(f_R(\vec{z})) - \text{NDCG}(g(\vec{z})) \right]^2 \cdot \pi_{\vec{x}}(\vec{z})$$

Here, $f_R$ is the original ranking model, $g$ is the surrogate model used for explanation, $\pi_{\vec{x}}$ is the KernelSHAP weighting kernel, $Z$ is the set of all possible coalitions (subsets of features), $\tau$ is Kendall's tau rank correlation coefficient, and ApproxNDCG is an approximation of the Normalized Discounted Cumulative Gain (NDCG).

## E.1   ANALYSIS OF RANKLIME

**Efficiency.**   The Efficiency axiom requires that the total attribution should equal the overall difference between the model output with all features and that with none. In RANKLIME, the loss minimizes an approximation error measured by `ApproxNDCG`. However, `ApproxNDCG` is not an additive measure. In other words, even if one could decompose NDCG over individual documents or features, its *approximation* may not retain this additivity.

*Intuitive Counter Example:* Suppose that for a given coalition the true NDCG decomposes as

$$\text{NDCG} = \sum_j g(rel_j) \cdot h(j).$$

If the approximation introduces nonlinearities (e.g., due to smoothing or rounding), then for two features $i$ and $j$, the marginal effects

$$\Delta_i \text{ApproxNDCG} \quad \text{and} \quad \Delta_j \text{ApproxNDCG}$$

need not sum to the total change. Thus, even if feature $i$ and feature $j$ are the only contributors, one might have

$$\Delta_i \text{ApproxNDCG} + \Delta_j \text{ApproxNDCG} \neq \text{Total change},$$

thereby violating Efficiency.

**Symmetry.**   Symmetry requires that if two features yield the same change in the model output in all coalitions, they should have equal attributions. However, the approximation in `ApproxNDCG` can be sensitive to ranking positions.

*Intuitive Counter Example:* Consider two features, $a$ and $b$, which individually yield identical changes in relevance scores. In a given coalition, due to subtle ranking reordering, the approximation may assign a slightly higher weight to the contribution of $a$ than to $b$ (or vice versa), because the ranking order determines the discount factors. Thus, even though $a$ and $b$ are symmetric in effect, the approximate metric may break the symmetry, leading to different attributions.

**Dummy (Missingness).**   For a feature that does not affect the model output (a dummy feature), the Shapley axiom requires that its contribution be zero. In RANKLIME, even a non-influential feature might change the ordering in the surrogate model (due to interactions with other features or the way missing features are treated in the approximation).

*Intuitive Counter Example:* If feature $c$ does not change any document's relevance in $f_R$, one would expect its contribution to vanish. However, when $c$ is added to a coalition, the computation of `ApproxNDCG` might slightly alter the ranking order (because the algorithm may treat missing features in a nontrivial manner), resulting in a small non-zero attribution. Hence, the Dummy axiom may be violated.

**Monotonicity.** Monotonicity implies that if the marginal contribution of a feature increases in every coalition, its attribution should not decrease. Since `ApproxNDCG` is a non-linear function of ranking order and not a linear combination of contributions, increases in the relevance of a feature do not translate in a straightforward way to increases in the approximate score.

*Intuitive Explanation:* Imagine a situation where increasing the relevance of a document moves it from a borderline position (with a high discount factor) to the top of the list, but due to the nonlinear nature of `ApproxNDCG`, the gain saturates. This means that further increases in relevance might have a negligible effect on the ApproxNDCG, even though the true marginal contribution should be larger. Consequently, the attributions computed by minimizing the loss may not increase monotonically, thereby violating the Monotonicity axiom.

### E.2 ANALYSIS OF RANKINGSHAP

**Efficiency.** RANKINGSHAP minimizes the squared difference of Kendall's tau ($\tau$) between the rankings of $f_R$ and $g$. Kendall's tau measures the rank correlation, not the absolute difference in scores. Thus, even if two coalitions have similar $\tau$ values, the overall difference in model output might not be captured.

*Intuitive Counter Example:* Suppose the full model output changes by a large amount when all features are present, but two different coalitions yield rankings that are highly correlated (i.e., high $\tau$). In this case, the loss may be small despite there being a significant change in the absolute values, meaning that the sum of the computed contributions (derived indirectly from $\tau$) does not match the total change. This discrepancy violates the Efficiency axiom.

**Symmetry.** Kendall's tau is based on pairwise comparisons of ranking orders. Two features that are symmetric in their true impact may, however, influence the ranking order differently due to interactions with other features. This can lead to different changes in $\tau$ when each is perturbed.

*Intuitive Explanation:* If features $a$ and $b$ are swapped in some coalitions, even though they are truly symmetric, the way $\tau$ counts pairwise disagreements might favor one ordering over the other. Thus, when minimizing the squared difference in $\tau$, the attributions computed for $a$ and $b$ might differ, thereby violating Symmetry.

**Dummy (Null Player).** A feature that does not affect the ranking should contribute zero to $\tau$. However, due to the permutation-based nature of Kendall's tau, even a dummy feature may change the number of concordant or discordant pairs in subtle ways.

*Intuitive Counter Example:* Imagine a feature $d$ that is irrelevant to the ranking. When $d$ is added to a coalition, if it is randomly inserted, the overall number of pairwise comparisons changes. Even though the feature does not affect the true ranking, this insertion can alter $\tau$ by a small amount, leading to a nonzero squared difference in the loss and thus a nonzero attribution.

**Monotonicity.** Since the loss in RANKINGSHAP is non-linear (squaring the difference in $\tau$), even if a feature's true marginal contribution increases, the resulting change in Kendall's tau may not increase proportionally.

*Intuitive Explanation:* Consider two coalitions where, in one, increasing the relevance of feature $e$ causes a small improvement in $\tau$, and in another, a larger improvement. If these changes do not align linearly with the changes in the model output (because $\tau$ is a rank statistic), the attribution derived from minimizing the squared differences might not reflect a monotonic increase in the true marginal effect. Hence, the Monotonicity axiom is not well approximated.

### E.3  ANALYSIS OF RANKSHAP

RANKSHAP uses the loss function

$$L_{\text{RANKSHAP}}(f_R, g, \pi_{\vec{x}}) = \sum_{\vec{z} \in Z} \Big[ \text{NDCG}\big(f_R(\vec{z})\big) - \text{NDCG}\big(g(\vec{z})\big) \Big]^2 \cdot \pi_{\vec{x}}(\vec{z}).$$

Because Normalized Discounted Cumulative Gain (NDCG) can be decomposed into a sum over documents, it admits an additive formulation:

$$\text{NDCG} = \sum_{j=1}^{n} g(rel_j) \cdot h(j),$$

where $g$ is a non-decreasing gain function and $h$ is a non-increasing discount function. This additivity is precisely what is needed for the Shapley axioms.

**Efficiency.**  Since NDCG is additive, the difference in NDCG between two coalitions is the sum of the individual differences. Thus, the attributions computed by RANKSHAP naturally sum to the total difference in the model outputs, satisfying the Efficiency axiom.

*Intuitive Proof:* If we denote the individual contribution of document $j$ by $g(rel_j) \cdot h(j)$, then the total effect of including a set of features is the sum over these contributions. The loss function directly minimizes the squared error between the true and surrogate NDCG scores. In doing so, it forces the surrogate model to distribute the total difference (the "explanation budget") among features in a way that respects additivity, which is the core idea behind the Shapley value.

**Symmetry.**  When two features affect the ranking in an identical manner, the corresponding changes in each document's gain function are the same. Since NDCG is additive and each document's contribution is determined independently by its position, symmetric features will receive equal contributions. Thus, the Symmetry axiom is satisfied.

**Dummy (Null Player).**  If a feature is dummy (i.e., it has no impact on any document's relevance), then its inclusion does not change the per-document gain values. Consequently, its marginal contribution to NDCG is zero, and the corresponding attribution will be zero, in agreement with the Dummy axiom.

**Monotonicity.**  Because an increase in the relevance score of any document leads to an increase in its gain $g(rel_j)$ and thus in its weighted contribution $g(rel_j) \cdot h(j)$, the additivity of NDCG ensures that the surrogate model's attributions increase monotonically with the true marginal contributions. This is in line with the Monotonicity axiom.

### E.4  CONCLUSION

- **RANKLIME:** The use of `ApproxNDCG` leads to a loss function that is not fully additive over features, and its sensitivity to ranking order may produce counter-intuitive attributions. Intuitive counter examples show that Efficiency, Symmetry, Dummy, and Monotonicity can be violated.

- **RANKINGSHAP:** By relying on Kendall's tau, which is a rank correlation measure, the loss function fails to capture absolute changes and linear contributions. This results in potential violations of Efficiency (since the sum of contributions does not match the total change), as well as Symmetry, Dummy, and Monotonicity violations due to nonlinear and permutation-sensitive behavior.

- **RANKSHAP:** In contrast, by using NDCG—which can be decomposed additively into gain and discount components—RANKSHAP aligns well with the Shapley axioms. Intuitive reasoning shows that the additivity inherent in NDCG ensures that Efficiency, Symmetry, Dummy, and Monotonicity are satisfied.

Thus, only RANKSHAP, with its loss function based on the additive properties of NDCG, produces explanations that fully approximate the fundamental Shapley axioms, while RANKLIME and RANKINGSHAP fall short due to their reliance on non-additive, non-linear measures.

# F    RANK COMPARISON METRICS

The following rank comparison metrics are popularly used in literature :

1. **Kendall's Tau:** Measures the number of pairs of alternatives over which two rankings disagree. Equivalently, it is also the minimum number of swaps of adjacent alternatives required to convert one ranking into another.

2. **Spearman's Footrule Distance**: Measures the total displacement of all alternatives between two rankings, i.e., the sum of the absolute differences between their positions in two rankings.

3. **Maximum Displacement Distance (MD)**: Measures the maximum of the displacements of all alternatives between two rankings.

4. **Cayley Distance**: Measures the minimum number of swaps (not necessarily of adjacent alternatives) required to convert one ranking into another.

5. **Copeland's Method**: Copeland's method is an algorithm to assign scores to candidates in Ranked choice voting. Quoting Wikipedia, each voter is asked to rank candidates in order of preference. A candidate A is said to have majority preference over another candidate B if more voters prefer A to B than prefer B to A; if the numbers are equal then there is a preference tie. The Copeland score for a candidate is the number of other candidates over whom he or she has a majority preference plus half the number of candidates with whom he or she has a preference tie. The winner of the election under Copeland's method is the candidate with the highest Copeland score; under Condorcet's method this candidate wins only if he or she has the maximum possible score of n − 1 where n is the number of candidates.

# G  USER STUDY INSTANCE

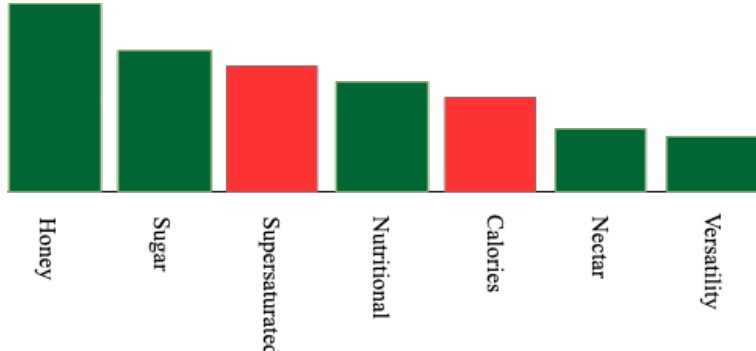

| |
|---|
| **P1** So while raw honey and sugar both contain glucose and fructose, our raw liquid gold is a nutrient dense food. Just as good grey sea salt is far healthier that iodized table salt, raw honey beats the buzz off of regular white sugar in both nutritional content and effects on the body. |
| **P2** Agave nectar has a taste and appearance similar to honey, making it a popular substitute for strict vegans and others who avoid honey. There are two kinds of agave nectar: dark agave nectar and light. Though the two can be used interchangeably, dark agave nectar has a stronger taste. |
| **P3** Clover honey is named so because it's produced from the nectar of clover blossoms. It has a mild flavor and sweetness and is favored in the United States for its abundance and versatility. When honey is labeled "pure," it means it has no additives such as sugar, corn syrup, or flavorings. |
| **P4** Energy Source. According to the USDA, honey contains about 64 calories per tablespoon. Therefore, it is used by many people as a source of energy. On the other hand, one tablespoon of sugar will give you about 15 calories. |
| **P5** 1 Honey crystallizes because it is a supersaturated solution. 2 This supersaturated state occurs because there is so much sugar in honey (more than 70%) relative to the water content (often less than 20%). |
| **Actual Query:** Is honey as a substitute for sugar healthier? |

Figure 1: A user study instance, depicting 5 passages and a corresponding feature attribution. Words that influenced the ranking decision positively are shown in green, whereas those that influenced negatively were shown in red. Participants were asked to re-rank documents based on the feature attribution to try and estimate the query.

**Question 2 : Read the following passages briefly**

1. The cheetah is capable of speed. up to 72 mph (114 km/h) and can maintain this speed over an average prey chase of 3.5 miles.. 32 m/s. The cheetah (Acinonyx jubatus) is one of the fastest mammals found in the animal kingdom today. The maximum speed of a cheetah is documented with the speed of about 30 m/s (70 mph).
2. While accelerating for time t c1, the cheetah travels a distance of. The cheetah then travels d c2 = 400 m, so that the total distance travelled by the cheetah is d c = 452.5 m. The gazelle starts at an initial speed v ig = 0 m/s and accelerates to a top speed of v fg = 70 km/h = 19.4 m/s in time.he fastest land mammal, the cheetah, is able to accelerate from a standing start to 96 km/h in just three seconds, which corresponds to an acceleration of 8.9 m/s 2.
3. The cheetah can accelerate from 0 km/h to 96 km/h in 3 seconds, this gives an acceleration of a c = 8.9 m/s 2. It has a top speed of v fc = 100 km/h = 30.6 m/s.The time to reach its top speed from rest, v ic = 0, is. The time t c2 to travel at top speed v fc, for a distance d c2 = 400 m is given by.he fastest land mammal, the cheetah, is able to accelerate from a standing start to 96 km/h in just three seconds, which corresponds to an acceleration of 8.9 m/s 2.
4. EMG or Electromyography is a test that is used to study the nerve and muscle function. EMG testing can provide your doctor with information about the extent and type of nerve and muscle disease/injury and can also determine the location of injury and give some indication whether the damage is reversible.
5. It is the same. paper the monitor normally prints on, so all of the same factors apply. o Small squares = 1 mm in height and 1 mm in width. o Large squares = 5 mm in height and 5 mm in width. You require this knowledge because when assessing the 12-lead ECG you will be.

Figure 2: User Study Passage Snapshot

## H  USER STUDY DETAILS

### H.1  ANONYMIZED INSTRUCTIONS

For each question, you will be shown 5 short passages related to a web search query in no particular order, and a bar chart of important words as explanation depicting an unknown ranking model's rationale for ordering those answers. You have to briefly read the passages and then using your intuition (i) reorder the passages from most relevant to least relevant based on the explanation (ii) guess the query/question that led to this particular ordering.

A word in the bar chart explanation, with positive or negative values, indicate whether they affected the order of documents that contain them, positively or negatively. Your response, which would be of the form (3,1,2,5,4) and a short text phrase/question would be recorded. This would just be used to understand how well the explanations aid in human understanding of an unknown model's intent.

This session has 10 such questions and should take no longer than 30 minutes. We don't recommend spending more than 3 minutes on a question. You would be completed within Cash cards worth $10 on successful completion of a session.

### H.2  USER STUDY SNAPSHOT

Snapshots from the User study for a particular query-document set, as seen by study participants are shown in Figures 2,3. The study was conducted via Google Forms.

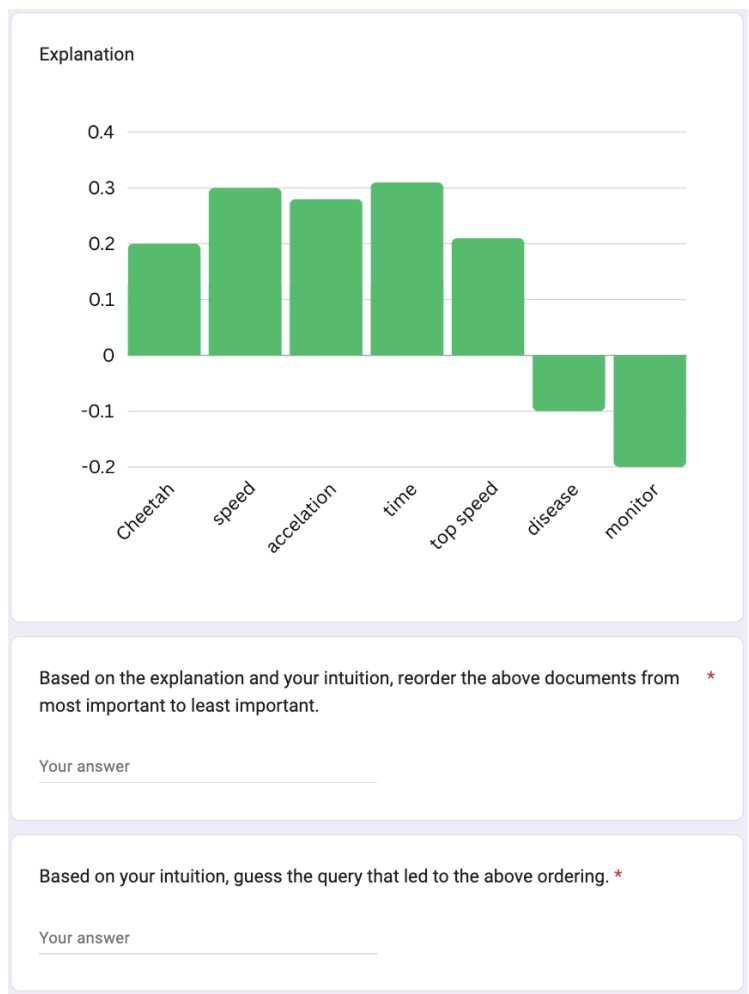

Figure 3: User Study Bar Chart and questions snapshot

# I  RANKSHAP PERFORMANCE WITH DIFFERENT VALUE FUNCTIONS

We run experiments comparing the performance of the RankSHAP framework with different value functions, namely Mean Average Precision (MAP), Cumulative Gain (CG), Discounted Cumulative Gain (DCG) and Normalized Discounted Cumulative Gain (NDCG) in Table 4. Below we discuss the same.

## I.1  COMPARATIVE ANALYSIS OF RANKSHAP VARIANTS

RankSHAP, employing different value functions (CG, MAP, DCG, and NDCG), exhibits clear performance distinctions across datasets, rankers, and query-document configurations. Among these, RankSHAP(NDCG) consistently achieves the highest fidelity and weighted fidelity (wFidelity) scores, outperforming the other RankSHAP variants. Aggregating across all configurations, RankSHAP(NDCG) delivers an average fidelity improvement of **8.3%** over RankSHAP(DCG), **11.5%** over RankSHAP(MAP), and **13.8%** over RankSHAP(CG). The performance edge of NDCG stems from its logarithmic discounting of ranks, which effectively prioritizes highly relevant documents, making it particularly suited for tasks where top-ranked items have a disproportionate impact on the evaluation.

Interestingly, the gains achieved by RankSHAP(NDCG) are most pronounced in the Top-10 setting, where it surpasses RankSHAP(CG) by an average of **17.4%** in fidelity and **15.8%** in wFidelity. In contrast, the improvements diminish slightly for Top-100, where NDCG maintains an edge of approximately **7.9%** over DCG and **10.4%** over MAP. These trends suggest that while all RankSHAP variants benefit from the Shapley-based attribution framework, those incorporating normalized or discounted value functions, like NDCG and DCG, are better suited for ranking tasks that emphasize relevance. Meanwhile, CG and MAP exhibit competitive yet slightly lower fidelity scores, likely because they lack the nuanced scaling necessary to capture differences in document importance effectively. Overall, RankSHAP(NDCG) emerges as the most robust and versatile variant, maintaining high fidelity across diverse scenarios.

## I.2  COMPARISON OF RANKSHAP VARIANTS WITH BASELINES

When compared to baseline methods such as Random, EXS, RankLIME, and RankingSHAP, all RankSHAP variants demonstrate significantly higher fidelity and weighted fidelity (wFidelity) scores across rankers and datasets. Among the baselines, RankingSHAP generally performs better, but RankSHAP(NDCG) consistently outperforms it with an average fidelity gain of **16.7%** across all settings, highlighting the advantage of incorporating axiomatic value functions like NDCG into the Shapley framework. For instance, in the Top-10 setting with the BM25 ranker on the MS MARCO dataset, RankSHAP(NDCG) achieves fidelity scores that are **21.2%** higher than RankingSHAP, showcasing its superior ability to identify and attribute relevant features accurately.

# J  RANKSHAP WITH EXPLICIT VS IMPLICIT RELEVANCE JUDGEMENTS

In scenarios where absolute relevance judgments are not available, we propose generating synthetic relevance labels based on predefined rules or heuristic relevance functions. This approach involves leveraging document features such as term frequency, topic overlap, or query-document embeddings to approximate relevance proxies. Specifically, we recommend using BM25, a widely adopted baseline relevance measure in information retrieval tasks. BM25 is computationally efficient and scales well for large document collections. It effectively balances term frequency (TF) saturation and inverse document frequency (IDF), ensuring robust performance across varying term distributions.

To evaluate the impact of using BM25 as a relevance proxy, we rerun certain RankSHAP experiments by comparing results obtained with exact relevance measures to those derived using BM25. We run experiments for BERT and T5 ranking models on the MS MARCO datasets. We report fidelity and weighted fidelity (wFidelity) scores across four ranking value functions: MAP, CG, DCG, and NDCG.

The results of our study can be found in Table 5. For both models, explicit relevance scores consistently yield higher fidelity and weighted fidelity (wFidelity) across all experimental settings (top-10, top-20, top-100). The performance drop when switching from explicit to BM25-based scores is modest, with fidelity reductions ranging between $6\%$ and $14\%$, depending on the value function and the number of top documents. Notably, the drop is more pronounced for smaller document sets (e.g., top-10), where precise relevance judgments likely have a greater impact on the accuracy of feature attributions.

With the top-10 documents, the fidelity of RankSHAP(NDCG) drops from 0.59 (explicit) to 0.54 (BM25) for BERT, and from 0.56 (explicit) to 0.52 (BM25) for T5. These results suggest that BM25 serves as a reasonably effective proxy for relevance, particularly when explicit scores are unavailable, albeit with slight performance degradation. Overall, the findings demonstrate that RankSHAP remains robust with implicit relevance scores, though explicit scores offer a tangible advantage, particularly in scenarios requiring finer-grained fidelity measures.

It is important to note that a limitation of BM25 is its reliance on exact keyword matches, which can overlook semantic relationships such as synonyms or paraphrases.

Table 4: Comparing performance between different feature attribution methods (Random, EXS, RankLIME, RankingSHAP and RankSHAP) for token-based explanations of textual rankers (BM25, BERT, T5 and LLAMA2) based on model fidelity (Fidelity) and weighted-fidelity (wFidelity) with ranking models fine tuned on the MS MARCO (MM) and TREC ROBUST 2004 (R04) datasets. Within RankSHAP, we generate feature attributions using 4 different value functions : Cumulative Gain (CG), Mean Average Precision (MAP), Discounted Cumulative Gain (DCG) and Normalized Discounted Cumulative Gain (NDCG). For these experiments, we generate feature attributions for each ranking models with the 10, 20 and 100 most relevant documents for a particular query respectively.

| Ranker ↓ | | Top-10 | | Top-20 | | Top-100 | |
|---|---|---|---|---|---|---|---|
| Metric → | | Fidelity | wFidelity | Fidelity | wFidelity | Fidelity | wFidelity |
| MM/BM25 | Random | 0.08 | 0.04 | -0.05 | 0.01 | -0.09 | 0.00 |
| | EXS | 0.39 | 0.34 | 0.31 | 0.29 | 0.24 | 0.17 |
| | RankLIME | 0.45 | 0.37 | 0.37 | 0.29 | 0.26 | 0.20 |
| | RankingSHAP | 0.52 | 0.41 | 0.45 | 0.33 | 0.31 | 0.24 |
| | **RankSHAP(MAP)** | 0.58 | 0.41 | 0.50 | 0.36 | 0.41 | 0.29 |
| | **RankSHAP(CG)** | 0.53 | 0.38 | 0.47 | 0.34 | 0.39 | 0.26 |
| | **RankSHAP(DCG)** | 0.60 | 0.42 | 0.51 | 0.38 | 0.42 | 0.31 |
| | **RankSHAP(NDCG)** | **0.63** | **0.48** | **0.54** | **0.40** | **0.47** | **0.33** |
| MM/BERT | Random | 0.12 | 0.02 | -0.09 | 0.01 | -0.12 | 0.00 |
| | EXS | 0.36 | 0.21 | 0.28 | 0.18 | 0.11 | 0.08 |
| | RankLIME | 0.41 | 0.24 | 0.32 | 0.24 | 0.23 | 0.18 |
| | RankingSHAP | 0.45 | 0.29 | 0.38 | 0.27 | 0.28 | 0.24 |
| | **RankSHAP(MAP)** | 0.54 | 0.42 | 0.38 | 0.36 | 0.33 | 0.26 |
| | **RankSHAP(CG)** | 0.49 | 0.48 | 0.35 | 0.29 | 0.28 | 0.24 |
| | **RankSHAP(DCG)** | 0.56 | 0.43 | 0.40 | 0.34 | 0.35 | 0.28 |
| | **RankSHAP(NDCG)** | **0.59** | **0.41** | **0.45** | **0.38** | **0.39** | **0.29** |
| MM/T5 | Random | 0.05 | 0.03 | -0.11 | 0.01 | -0.04 | 0.00 |
| | EXS | 0.36 | 0.21 | 0.30 | 0.17 | 0.11 | 0.06 |
| | RankLIME | 0.35 | 0.24 | 0.35 | 0.20 | 0.27 | 0.17 |
| | RankingSHAP | 0.41 | 0.30 | 0.38 | 0.32 | 0.31 | 0.22 |
| | **RankSHAP(MAP)** | 0.50 | 0.34 | 0.40 | 0.33 | 0.32 | 0.24 |
| | **RankSHAP(CG)** | 0.44 | 0.32 | 0.38 | 0.31 | 0.30 | 0.20 |
| | **RankSHAP(DCG)** | 0.55 | 0.38 | **0.43** | 0.35 | 0.34 | 0.27 |
| | **RankSHAP(NDCG)** | **0.56** | **0.39** | **0.43** | **0.37** | **0.36** | **0.29** |
| MM/LLAMA2 | Random | 0.04 | 0.01 | -0.2 | 0.0 | 0.02 | 0.1 |
| | EXS | 0.39 | 0.20 | 0.32 | 0.15 | 0.17 | 0.05 |
| | RankLIME | 0.35 | 0.24 | 0.35 | 0.20 | 0.27 | 0.17 |
| | RankingSHAP | 0.44 | 0.33 | 0.37 | 0.23 | 0.31 | 0.22 |
| | **RankSHAP(MAP)** | 0.55 | 0.38 | 0.40 | 0.33 | 0.32 | 0.25 |
| | **RankSHAP(CG)** | 0.48 | 0.37 | 0.36 | 0.27 | 0.31 | 0.24 |
| | **RankSHAP(DCG)** | 0.58 | 0.40 | 0.43 | 0.36 | 0.36 | 0.28 |
| | **RankSHAP(NDCG)** | **0.60** | **0.42** | **0.45** | **0.39** | **0.39** | **0.32** |
| R04/BM25 | Random | 0.08 | 0.04 | -0.04 | 0.00 | -0.07 | 0.00 |
| | EXS | 0.37 | 0.33 | 0.30 | 0.26 | 0.22 | 0.18 |
| | RankLIME | 0.43 | 0.36 | 0.36 | 0.25 | 0.25 | 0.18 |
| | RankingSHAP | 0.45 | 0.35 | 0.37 | 0.29 | 0.26 | 0.24 |
| | **RankSHAP(MAP)** | 0.51 | 0.37 | 0.44 | 0.33 | 0.32 | 0.26 |
| | **RankSHAP(CG)** | 0.48 | 0.35 | 0.36 | 0.30 | 0.25 | 0.25 |
| | **RankSHAP(DCG)** | 0.59 | 0.42 | **0.52** | 0.36 | 0.42 | 0.29 |
| | **RankSHAP(NDCG)** | **0.61** | **0.44** | **0.52** | **0.38** | **0.45** | **0.31** |
| R04/BERT | Random | 0.12 | 0.02 | -0.09 | 0.01 | -0.12 | 0.00 |
| | EXS | 0.35 | 0.20 | 0.27 | 0.16 | 0.10 | 0.07 |
| | RankLIME | 0.40 | 0.22 | 0.31 | 0.23 | 0.23 | 0.17 |
| | RankingSHAP | 0.43 | 0.26 | 0.36 | 0.28 | 0.28 | 0.23 |
| | **RankSHAP(MAP)** | 0.48 | 0.35 | 0.40 | 0.34 | 0.37 | 0.24 |
| | **RankSHAP(CG)** | 0.44 | 0.28 | 0.36 | 0.28 | 0.29 | 0.21 |
| | **RankSHAP(DCG)** | 0.56 | **0.41** | 0.42 | 0.35 | 0.37 | 0.26 |
| | **RankSHAP(NDCG)** | **0.59** | **0.41** | **0.45** | **0.38** | **0.39** | **0.29** |
| R04/T5 | Random | 0.05 | 0.03 | -0.11 | 0.01 | -0.04 | 0.00 |
| | EXS | 0.38 | 0.21 | 0.32 | 0.17 | 0.13 | 0.06 |
| | RankLIME | 0.38 | 0.24 | 0.35 | 0.20 | 0.28 | 0.17 |
| | RankingSHAP | 0.41 | 0.31 | 0.38 | 0.24 | 0.32 | 0.24 |
| | **RankSHAP(MAP)** | 0.49 | 0.34 | 0.40 | 0.27 | 0.34 | 0.24 |
| | **RankSHAP(CG)** | 0.42 | 0.30 | 0.39 | 0.24 | 0.33 | 0.24 |
| | **RankSHAP(DCG)** | 0.54 | 0.37 | 0.40 | 0.33 | 0.35 | 0.25 |
| | **RankSHAP(NDCG)** | **0.56** | **0.39** | **0.43** | **0.37** | **0.36** | **0.29** |

Table 5: Comparing performance between different feature attribution methods (Random, EXS, RankLIME, RankingSHAP) and RankSHAP (with differnt value functions and different methods to model relevance) for token-based explanations of textual rankers (BERT and T5) based on model fidelity (Fidelity) and weighted-fidelity (wFidelity) with ranking models fine tuned on the MS MARCO (MM) dataset. Within RankSHAP, we generate feature attributions using 4 different rank value functions : Cumulative Gain (CG), Mean Average Precision (MAP), Discounted Cumulative Gain (DCG) and Normalized Discounted Cumulative Gain (NDCG) and 2 different methods to compute relevance (Explicit ranking model score vs BM25 score). For these experiments, we generate feature attributions for each ranking models with the 10, 20 and 100 most relevant documents for a particular query respectively.

| Ranker ↓ | | Top-10 | | Top-20 | | Top-100 | |
|---|---|---|---|---|---|---|---|
| Metric → | | Fidelity | wFidelity | Fidelity | wFidelity | Fidelity | wFidelity |
| MM/BERT | Random | 0.12 | 0.02 | -0.09 | 0.01 | -0.12 | 0.00 |
| | EXS | 0.36 | 0.21 | 0.28 | 0.18 | 0.11 | 0.08 |
| | RankLIME | 0.41 | 0.24 | 0.32 | 0.24 | 0.23 | 0.18 |
| | RankingSHAP | 0.45 | 0.29 | 0.38 | 0.27 | 0.28 | 0.24 |
| | RankSHAP(BM25/MAP) | 0.47 | 0.32 | 0.31 | 0.33 | 0.28 | 0.22 |
| | RankSHAP(BM25/CG) | 0.44 | 0.43 | 0.31 | 0.26 | 0.24 | 0.21 |
| | RankSHAP(BM25/DCG) | 0.50 | 0.37 | 0.36 | 0.30 | 0.31 | 0.25 |
| | RankSHAP(BM25/NDCG) | 0.54 | 0.38 | 0.42 | 0.37 | 0.36 | 0.25 |
| | RankSHAP(MAP) | 0.54 | 0.42 | 0.38 | 0.36 | 0.33 | 0.26 |
| | RankSHAP(CG) | 0.49 | 0.48 | 0.35 | 0.29 | 0.28 | 0.24 |
| | RankSHAP(DCG) | 0.56 | 0.43 | 0.40 | 0.34 | 0.35 | 0.28 |
| | RankSHAP(NDCG) | **0.59** | **0.41** | **0.45** | **0.38** | **0.39** | **0.29** |
| MM/T5 | Random | 0.05 | 0.03 | -0.11 | 0.01 | -0.04 | 0.00 |
| | EXS | 0.36 | 0.21 | 0.30 | 0.17 | 0.11 | 0.06 |
| | RankLIME | 0.35 | 0.24 | 0.35 | 0.20 | 0.27 | 0.17 |
| | RankingSHAP | 0.41 | 0.30 | 0.38 | 0.32 | 0.31 | 0.22 |
| | RankSHAP(BM25/MAP) | 0.45 | 0.30 | 0.37 | 0.29 | 0.30 | 0.21 |
| | RankSHAP(BM25/CG) | 0.41 | 0.28 | 0.36 | 0.27 | 0.27 | 0.18 |
| | RankSHAP(BM25/DCG) | 0.51 | 0.35 | 0.42 | 0.31 | 0.31 | 0.24 |
| | RankSHAP(BM25/NDCG) | 0.52 | 0.36 | 0.40 | 0.34 | 0.32 | 0.27 |
| | RankSHAP(MAP) | 0.50 | 0.34 | 0.40 | 0.33 | 0.32 | 0.24 |
| | RankSHAP(CG) | 0.44 | 0.32 | 0.38 | 0.31 | 0.30 | 0.20 |
| | RankSHAP(DCG) | 0.55 | 0.38 | **0.43** | 0.35 | 0.34 | 0.27 |
| | RankSHAP(NDCG) | **0.56** | **0.39** | **0.43** | **0.37** | **0.36** | **0.29** |

