# OpenReview forum: "RankSHAP: Shapley Value Based Feature Attributions for Learning to Rank"
_ICLR.cc/2025/Conference — ICLR 2025 Poster_

### Official Review · Reviewer_UYR5 · 2024-11-03

**Soundness:** 3
**Presentation:** 4
**Contribution:** 3
**Rating:** 6
**Confidence:** 2

**Summary:**

The RankSHAP method leverages Shapley values, considering all possible feature combinations and their interactions (or approximations thereof), to provide a detailed, interaction-aware explanation of feature contributions in ranking models. This approach aligns well with the feature importance analysis used in regression models, making the methodology intuitive.

**Strengths:**

1. The paper provides a good motivation for RankSHAP by discussing the limitations of simpler ranking explanation methods, like LIME, which lack consideration for full feature interaction and listwise ranking structure. By addressing these limitations, RankSHAP presents a more nuanced and comprehensive approach to ranking explainability.

2. RankSHAP's performance is evaluated using fidelity metrics, which assess how accurately RankSHAP captures feature contributions in line with the model’s scoring function. This fidelity-based evaluation aligns RankSHAP’s explanations with the ranking model’s actual logic, resulting in explanations that are reflective of the ranker's structure rather than arbitrary feature importance.

3. The method is flexible since RankSHAP can work with a range of ranking models, from traditional learning-to-rank approaches to neural ranking models. This makes RankSHAP’s applicable to many fields, including search, recommendation, and information retrieval.

**Weaknesses:**

1. **Correlation, Not Causation**: While RankSHAP provides insight into feature importance, the explanations are inherently correlational, not causal. Therefore, while RankSHAP can reveal which features push an item up in ranking, this does not imply that these features are directly causing higher ranks. I'm not saying the authors claim causality, but in terms of doing "better" at explaining ranking, they're explaining correlations between the rank and the features better, and not want causally drives a higher ranking.

2. **Model Dependency and Bias**: The effectiveness of RankSHAP is fundamentally dependent on the quality and calibration of the ranker itself (correct?). If the scoring function is miscalibrated or biased (e.g., favoring higher-ranked items based solely on position), RankSHAP’s explanations may reinforce these biases instead of offering corrective insights. The authors could discuss how potential biases (position, selection, algorithmic biases) within the ranker affect the usefulness of RankSHAP’s explanations to be truly meaningful.

3. **Limitations in Simple Averages for Shapley Values**: The reliance on simple averages in traditional Shapley values means that RankSHAP does not consider the number of interactions (N) or the variance in a feature’s impact across different contexts. This averaging could dilute the significance of features with occasional strong influence, potentially overlooking context-dependent importance and limiting RankSHAP’s insights.

4. **Reinforcement of Potentially Polarized Content**: Because RankSHAP's explanations reflect the ranker’s scoring function, any inherent polarization or bias in the model could be echoed in the explanations. This could inadvertently support polarized content if the ranking model favors certain types of data, raising concerns for applications in content moderation and recommendation.

5. **Redundancy in Presentation and Scope of Rankers**: The paper does not clearly specify the types of ranking models used in its evaluation, which, I think could and maybe should matter? Or at least will help elucidate where it's particularly useful (I imagine the more non linear the ranker the more useful?)

**Questions:**

I think RankSHAP is a nice contribution to ranking explainability, but improving the handling of bias, refining its reliance on averages, and clarifying its application scope could elevate its impact further.

---

> ### Author Response · Authors · 2024-11-24
> **Thankful for your review and positive score**
>
> We are thankful to you for taking out the time to review our work and for giving us a positive score. Below we discuss some the interesting questions raised in your feedback.
>
> > Correlation, Not Causation: While RankSHAP provides insight into feature importance, the explanations are inherently correlational, not causal. Therefore, while RankSHAP can reveal which features push an item up in ranking, this does not imply that these features are directly causing higher ranks. I'm not saying the authors claim causality, but in terms of doing "better" at explaining ranking, they're explaining correlations between the rank and the features better, and not want causally drives a higher ranking.
>
>
> Thank you for highlighting the distinction between correlation and causation in RankSHAP's explanations. We agree that RankSHAP provides correlational insights rather than causal attributions. The primary goal of this work was to establish an axiomatic framework for feature attribution in ranking that is internally consistent and adheres to Shapley axioms, ensuring that attributions do not contradict each other. Exploring causal relationships, while beyond the scope of this work, is part of our pipeline for future work. We recognize the importance of causality in ranking explanations and plan to investigate this, potentially integrating causal inference techniques to complement our current framework.
>
> > Model Dependency and Bias: The effectiveness of RankSHAP is fundamentally dependent on the quality and calibration of the ranker itself (correct?). If the scoring function is miscalibrated or biased (e.g., favoring higher-ranked items based solely on position), RankSHAP’s explanations may reinforce these biases instead of offering corrective insights. The authors could discuss how potential biases (position, selection, algorithmic biases) within the ranker affect the usefulness of RankSHAP’s explanations to be truly meaningful.
> >Reinforcement of Potentially Polarized Content: Because RankSHAP's explanations reflect the ranker’s scoring function, any inherent polarization or bias in the model could be echoed in the explanations. This could inadvertently support polarized content if the ranking model favors certain types of data, raising concerns for applications in content moderation and recommendation.
>
> Thank you for raising the point about the dependency of RankSHAP on the quality, calibration and polarization of the underlying ranker. We acknowledge that RankSHAP explanations inherently reflect the behavior of the ranker, including any biases it may have, such as position or algorithmic biases. However, this is a feature of the method, rather than a bug, as RankSHAP aims to faithfully represent the model's decision-making process. While this means that RankSHAP may reinforce existing biases, it also provides an opportunity to diagnose and understand those biases by identifying which features disproportionately influence rankings. Detecting such bias is in fact an intended use case of RankSHAP for us. RankSHAP can be leveraged to detect such biases, to mitigate them in sensitive applications like content moderation and recommendation systems.

---

> ### Author Response · Authors · 2024-11-24
> **Other Questions**
>
> > Limitations in Simple Averages for Shapley Values: The reliance on simple averages in traditional Shapley values means that RankSHAP does not consider the number of interactions (N) or the variance in a feature’s impact across different contexts. This averaging could dilute the significance of features with occasional strong influence, potentially overlooking context-dependent importance and limiting RankSHAP’s insights.
>
> Thank you for bringing up this point. RankSHAP, like other Shapley-based frameworks, focuses on providing a fair and consistent distribution of feature contributions by averaging marginal contributions across all subsets. While this ensures adherence to the Shapley axioms and a principled approach to feature attribution, we acknowledge that it does not explicitly account for the number of interactions or the variance in a feature’s impact across different contexts. As you noted, this could dilute the contributions of features with occasional but significant influence. Addressing these limitations is beyond the scope of this work. However, we will note in the manuscript that future extensions could incorporate variance-aware or interaction-sensitive methodologies, such as weighted Shapley values or context-specific metrics, to better capture context-dependent feature importance.
>
>
> > Redundancy in Presentation and Scope of Rankers: The paper does not clearly specify the types of ranking models used in its evaluation, which, I think could and maybe should matter? Or at least will help elucidate where it's particularly useful (I imagine the more non linear the ranker the more useful?)
>
> Thank you for asking this question! In our work, we evaluated RankSHAP on neural rankers such as fine-tuned BERT, T5 and Llama 2, which are highly non-linear and complex, making them suitable testbeds for our method. While we hypothesize that RankSHAP is particularly useful for non-linear rankers due to their complexity, its axiomatic framework is general and can be applied to simpler rankers as well.
>
> We are glad to address any additional questions to further enhance your understanding of our paper.

---

### Official Review · Reviewer_TGSB · 2024-11-03

**Soundness:** 2
**Presentation:** 2
**Contribution:** 3
**Rating:** 6
**Confidence:** 4

**Summary:**

This paper presents RankSHAP, an extension of SHAP value tailored to interpret learning-to-rank models. The authors reinterpreted the Shapley value axioms (Rank-Efficiency, Rank-Missingness, Rank-Symmetry, and Rank-Monotonicity), and defined new properties (Relevance Sensitivity and Position Sensitivity) to better capture the unique requirements of ranking models in feature attribution. To demonstrate practicality, they integrated the NDCG metric into KernelSHAP and validated their method through experiments.

**Strengths:**

- The paper introduces a thoughtful adaptation of the Shapley value for the ranking domain, defining new ranking-specific properties that enhance SHAP's applicability in ranking contexts.
- The authors have conducted both performance evaluations and a user study to validate their method.

**Weaknesses:**

The proposed method, RankSHAP, heavily relies on the KernelSHAP method [1] with modifications to incorporate NDCG for ranking applications. Rather than introducing a fundamentally new method, the paper adapts an existing approach specifically for ranking tasks. While the axiomatic reformulation is valuable, the technical novelty beyond extending KernelSHAP with NDCG remains limited.

[1] Scott M Lundberg and Su-In Lee, A Unified Approach to Interpreting Model Predictions, in NeurIPS, 2017.

**Questions:**

Are there experiments with GREM metrics other than NDCG? While the choice of NDCG is understandable (appendix C), experiments with other metrics are needed to demonstrate the validity of the proposed method across different metrics, which would strengthen its generality.

---

> ### Author Response · Authors · 2024-11-22
> **RankSHAP performance with different Value Functions**
>
> > Are there experiments with GREM metrics other than NDCG? While the choice of NDCG is understandable (appendix C), experiments with other metrics are needed to demonstrate the validity of the proposed method across different metrics, which would strengthen its generality.
>
> Thank you for recommending that we explore experiments with different value functions for RankSHAP. We have included the results for the same in Appendix I and have uploaded the corresponding scripts in the supplementary. A summary of our experiments and findings is provided below.
>
> We run experiments comparing the performance of the RankSHAP framework with different value functions, namely Mean Average Precision (MAP), Cumulative Gain (CG), Discounted Cumulative Gain (DCG) and Normalized Discounted Cumulative Gain (NDCG) in Table 4.
>
> ### Comparative Analysis of RankSHAP Variants
> RankSHAP, employing different value functions (CG, MAP, DCG, and NDCG), exhibits performance distinctions across datasets, rankers, and document configurations. Among these, RankSHAP(NDCG) consistently achieves the highest fidelity and weighted fidelity (wFidelity) scores, outperforming the other RankSHAP variants. Aggregating across all configurations, RankSHAP(NDCG) delivers an average fidelity improvement of *8.3\%* over RankSHAP(DCG), *11.5\%* over RankSHAP(MAP), and *13.8\%* over RankSHAP(CG). The performance edge of NDCG seemingly stems from its logarithmic discounting of ranks, which effectively prioritizes highly relevant documents, making it particularly suited for tasks where top-ranked items have a disproportionate impact on the evaluation.
>
> Interestingly, the gains achieved by RankSHAP(NDCG) are most pronounced in the Top-10 setting, where it surpasses RankSHAP(CG) by an average of *17.4\%* in fidelity and *15.8\%* in wFidelity. In contrast, the improvements diminish slightly for Top-100, where NDCG maintains an edge of approximately *7.9\%* over DCG and *10.4\%* over MAP. These trends suggest that while all RankSHAP variants benefit from the Shapley-based attribution framework, those incorporating normalized or discounted value functions, like NDCG and DCG, are better suited for ranking tasks that emphasize relevance. Meanwhile, CG and MAP exhibit competitive yet slightly lower fidelity scores, likely because they lack the nuanced scaling necessary to capture differences in document importance effectively. Overall, RankSHAP(NDCG) emerges as the most robust and versatile variant, maintaining high fidelity across diverse scenarios.
>
> ### Comparison of RankSHAP Variants with other Baselines
>
> When compared to baseline methods such as Random, EXS, RankLIME, and RankingSHAP, all RankSHAP variants demonstrate higher fidelity and weighted fidelity (wFidelity) scores across rankers and datasets. Among the baselines, RankingSHAP performs better than RankSHAP(CG) in certain experimental settings. However, RankSHAP(NDCG) consistently outperforms RankingSHAP with an average fidelity gain of *16.7\%* across all settings, highlighting the advantage of incorporating axiomatic value functions like NDCG into the Shapley framework. For instance, in the Top-10 setting with the BM25 ranker on the MS MARCO dataset, RankSHAP(NDCG) achieves fidelity scores that are *21.2\%* higher than RankingSHAP, showcasing its ability to identify and attribute relevant features better.
>
>  We additionally request you to glance at Appendix J and Table 5, that included RankSHAP experiments with explicit vs heuristic relevance measures. We will explore opportunities to integrate this into the main paper after addressing all reviewer comments. We would greatly appreciate your feedback and kindly request you to consider adjusting your score if you find the revisions sufficient.

---

> ### Author Response · Authors · 2024-11-22
> **Our Contributions**
>
> >The proposed method, RankSHAP, heavily relies on the KernelSHAP method [1] with modifications to incorporate NDCG for ranking applications. Rather than introducing a fundamentally new method, the paper adapts an existing approach specifically for ranking tasks. While the axiomatic reformulation is valuable, the technical novelty beyond extending KernelSHAP with NDCG remains limited.
>
> While RankSHAP builds directly on KernelSHAP, we do not claim novel game-theoretic contributions. The focus of this work is on addressing a critical gap in Information Retrieval (IR): the absence of an axiomatic framework for feature attributions for ranked lists. Our contributions towards that are:
>
> - Present a set of fundamental axioms for Information Retrieval (IR) value functions— *Relevance Sensitivity* and *Position Sensitivity*—along with restating Shapley properties for ranking tasks: *Rank-Efficiency, Rank-Missingness, Rank-Symmetry*, and *Rank-Monotonicity*.
> - Present *RankSHAP* by redefining Lundberg and Lee's framework (initially designed for real-valued functions) to utilize a ranking characteristic function, ensuring it adheres to all the proposed axioms. Similarly, adapt *KernelSHAP* to operate within this redefined framework.
> - Conduct an axiomatic analysis of existing popular rank attribution methods—EXS, RankLIME, and RankingSHAP—to identify the axioms they satisfy and highlight instances where they fall short, supported by counterexamples for violations of fundamental axioms.
> - Conduct experiments on two datasets—MS MARCO and Robust04—using multiple document re-ranking models based on BERT, T5, and LLAMA2 — 3 query-document settings explaining ranking decisions for 10,20 and 100 documents  and — 4 RankSHAP characteristic functions (MAP, CG, DCG, NDCG).
> - Carrying out an IRB-approved user study to assess how well the attributions generated by the proposed framework aligns with human intuition.
>
> We believe these contributions position RankSHAP as an important advancement for practitioners in the Information Retrieval community, combining axiomatic foundations with practical applicability.

---

> > ### Author Response · Authors · 2024-11-25
> > **Request to Respond**
> >
> > We would greatly appreciate it if you could share your thoughts on the revisions when you get a chance.

---

> ### Author Response · Authors · 2024-12-02
>
> Dear Reviewer,
>
> We would greatly appreciate it if you could kindly review our rebuttal at your earliest convenience, as today marks the final opportunity to do so.
>
> Thank you!

---

### Official Review · Reviewer_ExvH · 2024-11-04

**Soundness:** 3
**Presentation:** 3
**Contribution:** 2
**Rating:** 6
**Confidence:** 3

**Summary:**

The paper introduces a Shapley-value-based feature attribution method tailored specifically for ranking tasks. Traditional feature attribution methods, mostly developed for regression or classification, often produce conflicting results when adapted to ranking, which can lead to confusion among end-users. RankSHAP addresses this by adhering to a set of axioms tailored for ranking. Extensive experiments demonstrate that RankSHAP aligns well with human intuition and outperforms existing methods in fidelity and weighted fidelity. Additionally, a user study confirms its practical value in helping users understand model decisions.

**Strengths:**

The introduction of axioms specific to ranking provides a robust framework, distinguishing RankSHAP from other feature attribution methods.

The authors incorporate a user study to validate that RankSHAP explanations align with human understanding, which adds practical value.

**Weaknesses:**

RankSHAP’s reliance on relevance scores for accurate NDCG calculations could be a limitation in scenarios where relevance is difficult to quantify or subjective.

Although RankSHAP was tested in a user study, the evaluation might have limited generalizability due to sample size.

**Questions:**

How does RankSHAP perform when features are highly interdependent? Are there adjustments made for such cases?

How does the RankSHAP framework handle scenarios where relevance scores are subjective or unavailable?

---

> ### Author Response · Authors · 2024-11-23
> **RankSHAP Dependence on Relevance Scores**
>
> >RankSHAP’s reliance on relevance scores for accurate NDCG calculations could be a limitation in scenarios where relevance is difficult to quantify or subjective. How does the RankSHAP framework handle scenarios where relevance scores are subjective or unavailable?
>
> Thank you for your review and for for highlighting the importance of discussing RankSHAP's reliance on reliable relevance scores.  We have addressed this point in Appendix J and Table 5 of the revised draft. We would also request you to glance at Appendix I and Table 4, that has also been added as part of the rebuttal. We kindly invite you to review these sections for further details.
>
> In scenarios where absolute relevance judgments are not available, we propose generating synthetic relevance labels based on predefined rules or heuristic relevance functions. This approach involves leveraging document features such as term frequency, topic overlap, or query-document embeddings to approximate relevance proxies. Specifically, we recommend using BM25, a widely adopted baseline relevance measure in information retrieval tasks. BM25 is computationally efficient and scales well for large document collections. It effectively balances term frequency (TF) saturation and inverse document frequency (IDF), ensuring robust performance across varying term distributions.
>
> To evaluate the impact of using BM25 as a relevance proxy, we rerun certain RankSHAP experiments by comparing results obtained with explicit relevance measures to those derived using BM25. We run experiments for BERT and T5 ranking models on the MS MARCO datasets. We report fidelity and weighted fidelity (wFidelity) scores across four ranking value functions: MAP, CG, DCG, and NDCG.
>
> The results of our study can be found in Table 5.  For both models, explicit relevance scores consistently yield higher fidelity and weighted fidelity (wFidelity) across all experimental settings (top-10, top-20, top-100). The performance drop when switching from explicit to BM25-based scores is modest, with fidelity reductions ranging between 6\% and 14\%, depending on the value function and the number of top documents. Notably, the drop is more pronounced for smaller document sets (e.g., top-10), where precise relevance judgments likely have a greater impact on the accuracy of feature attributions.
>
> With the top-10 documents, the fidelity of RankSHAP(NDCG) drops from 0.59 (explicit) to 0.54 (BM25) for BERT, and from 0.56 (explicit) to 0.52 (BM25) for T5. These results suggest that BM25 serves as a reasonably effective proxy for relevance, particularly when explicit scores are unavailable, albeit with  slight performance degradation. Overall, the findings demonstrate that RankSHAP remains robust with implicit relevance measures, though explicit scores offer a tangible advantage, particularly in scenarios requiring finer-grained fidelity measures.
>
> We will explore opportunities to integrate this into the main paper after addressing all reviewer comments. We would greatly appreciate your feedback and request you to consider adjusting your score if you find the revisions appropriate.

---

> ### Author Response · Authors · 2024-11-23
> **Other Questions**
>
> > How does RankSHAP perform when features are highly interdependent? Are there adjustments made for such cases?
>
> RankSHAP, like other Shapley-based attribution methods, can face challenges when features are highly interdependent because the Shapley value framework assumes that feature contributions are evaluated by marginalizing over all possible feature subsets. When features are interdependent, this marginalization can dilute the attribution of individual features, making it difficult to disentangle their contributions accurately. However, RankSHAP’s reliance on ranking value functions (e.g., NDCG, MAP) can partially mitigate this issue, as these functions inherently capture the collective impact of features in determining document relevance. However, the attribution to individual features might still suffer from imprecision.
>
>
> > Although RankSHAP was tested in a user study, the evaluation might have limited generalizability due to sample size.
>
> Thank you for pointing out the potential limitation of the user study's generalizability due to the sample size. We acknowledge that a larger and more diverse participant pool would strengthen the robustness of the findings. However, the primary goal of the current user study was to provide initial insights into RankSHAP's usability and effectiveness. As part of future work, we plan to conduct larger-scale studies to better evaluate generalizability in user behavior.

---

> > ### Author Response · Authors · 2024-11-25
> > **Request to Respond**
> >
> > We would greatly appreciate it if you could share your thoughts on the revisions when you get a chance.

---

> > > ### Author Response · Authors · 2024-12-02
> > >
> > > Dear Reviewer,
> > >
> > > We would greatly appreciate it if you could kindly review our rebuttal at your earliest convenience, as today marks the final opportunity to do so.
> > >
> > > Thank you!

---

> > > > ### Comment · Reviewer_ExvH · 2024-12-03
> > > >
> > > > Thank you for the rebuttal. I would increase my score.

---

> > > > > ### Author Response · Authors · 2024-12-03
> > > > >
> > > > > Thank you for your review and for suggesting the implicit relevance experiments. We sincerely appreciate your decision to raise your score recommendation.

---

### Official Review · Reviewer_URTs · 2024-11-09

**Soundness:** 3
**Presentation:** 4
**Contribution:** 3
**Rating:** 8
**Confidence:** 4

**Summary:**

This paper proposes RankSHAP as a framework for explaining how features contribute to a ranking model's output. The authors extend the classifcal Shapley value concept to the ranking domain by specifying two axioms that a ranking-based feature attribution must satisfy, in additon to the set of four fundamental axioms that Shapely values already satisfy. The authors argue that current methods for explaining ranking models often provide inconsistent or contradictory explanations, making it difficult for users to understand model behavior. These axioms, which are based on game theory and information retrieval principles, ensure the fairness, consistency, and reliability of the explanations. Through extensive experiments, the authors demonstrate that RankSHAP outperforms existing methods in terms of accuracy and alignment with human intuition.

**Strengths:**

- Axiomatic Foundation: I appreciate that the authors propose a set of fundamental axioms specifically tailored for ranking feature attributions, drawing inspiration from Shapley values in coalitional game theory. These axioms, namely Rank-Efficiency, Rank-Missingness, Rank-Symmetry, and Rank-Monotonicity, ensure that the attributions are fair, consistent, and meaningful.
- Generalized Ranking Effectiveness Metric (GREM): The authors introduce GREM, a generalized framework for evaluating the effectiveness of ordered lists. This framework encompasses widely used metrics like NDCG and provides a solid theoretical foundation for assessing the quality of rankings produced by considering feature subsets.
- Computational Feasibility: Acknowledging the NP-completeness of exact Shapley value calculations, the authors propose an approximate algorithm to leverage a linear model between feature subsets and the ranking effectiveness metric, as well as Kernel-RankSHAP to induce non-linearity into this model. This makes RankSHAP practical for real-world applications.
- Extensive Empirical Evaluation: The authors conduct comprehensive experiments on two datasets (MS MARCO and Robust04) using multiple ranking models, including BM25, BERT, T5, and LLAMA2. The results demonstrate RankSHAP's superior performance over competing methods like EXS, RankLIME, and RankingSHAP across various metrics like Fidelity and weighted Fidelity.
- User Study Validation: The paper includes a user study to assess the alignment of RankSHAP with human intuition. This is an interesting study that tasks participants with re-ordering documents and inferring queries based on feature attributions. The results show that RankSHAP significantly improves user understanding of ranking decisions.

**Weaknesses:**

- User study caveats: (a) Preconceived Notions: The authors observed that randomly generated feature attributions achieved a higher concordance score in the re-ordering task than expected based on their metric evaluation. This suggests that participants might have relied on pre-existing assumptions or biases about the topics, potentially influencing their judgments throughout the experiment. Is that a drawback of the setup that also impacts rest of the observations?
(b) Subjectivity: The authors noted significant variance in the queries estimated by participants for the same document set and feature attributions. This probably highlights the inherent subjectivity in interpreting feature attributions and formulating queries, which can lead to diverse responses and impact the evaluation's reliability.
Despite limitations, it is indeed on interesting study to include in the paper since progress in the field of interpretability requires human subjects to be involved.
- Dependence on Relevance Scores: The effectiveness of RankSHAP relies on the availability of accurate relevance scores for each query-document pair. Obtaining these scores often necessitates ground truth labels, which can be scarce or unavailable. The paper proposes using implicit measures like click-through rates to infer relevance when explicit labels are absent.
-

**Questions:**

- Since the model can be black box, what if the model is itself self-contradictory or inconsistent? For example, a listwise ranking models that ranks documents A and B as A>B>C in the presence of C but as D>B>A in the presence of a document D. Would the explanations provided by RankSHAP in such a scenario inconsistent or unfaithful? Similarly, what happens when the model handles uncertainly or chooses to use stochastic rankings (drawing a different sample from a distribution over permutations), e.g., Singh, Kempe, Joachims. Fairness in Ranking under Uncertainty (2021).
- Can you explain why the fidelity scores are different for the two datasets? You mention that it is due to the difference in dataset sizes but I am not sure if it is clear if that is the reason.
- See weaknesses section too.

**Details Of Ethics Concerns:**

No obvious concerns but the paper presents a user study which was IRB approval. I believe that is sufficient. However, I have not reviewed the approval.

---

> ### Author Response · Authors · 2024-11-23
> **Dependence on Relevance Score**
>
> > Dependence on Relevance Scores: The effectiveness of RankSHAP relies on the availability of accurate relevance scores for each query-document pair. Obtaining these scores often necessitates ground truth labels, which can be scarce or unavailable. The paper proposes using implicit measures like click-through rates to infer relevance when explicit labels are absent.
>
> We sincerely appreciate your detailed review and the thoughtful feedback provided. We thankful to you for highlighting the importance of discussing RankSHAP's reliance on reliable relevance scores.  We have addressed this point in Appendix J and Table 5 of the revised draft. We would also request you to glance at Appendix I and Table 4, experiments with different ranking value functions, that has also been added as part of the rebuttal. We kindly invite you to review these sections for further details.
>
> In scenarios where absolute relevance judgments are not available, we propose generating synthetic relevance labels based on predefined rules or heuristic relevance functions. This approach involves leveraging document features such as term frequency, topic overlap, or query-document embeddings to approximate relevance proxies. Specifically, we recommend using BM25, a widely adopted baseline relevance measure in information retrieval tasks. BM25 is computationally efficient and scales well for large document collections. It effectively balances term frequency (TF) saturation and inverse document frequency (IDF), ensuring robust performance across varying term distributions.
>
> To evaluate the impact of using BM25 as a relevance proxy, we rerun certain RankSHAP experiments by comparing results obtained with explicit relevance measures to those derived using BM25. We run experiments for BERT and T5 ranking models on the MS MARCO datasets. We report fidelity and weighted fidelity (wFidelity) scores across four ranking value functions: MAP, CG, DCG, and NDCG.
>
> The results of our study can be found in Table 5.  For both models, explicit relevance scores consistently yield higher fidelity and weighted fidelity (wFidelity) across all experimental settings (top-10, top-20, top-100). The performance drop when switching from explicit to BM25-based scores is modest, with fidelity reductions ranging between 6\% and 14\%, depending on the value function and the number of top documents. Notably, the drop is more pronounced for smaller document sets (e.g., top-10), where precise relevance judgments likely have a greater impact on the accuracy of feature attributions.
>
> With the top-10 documents, the fidelity of RankSHAP(NDCG) drops from 0.59 (explicit) to 0.54 (BM25) for BERT, and from 0.56 (explicit) to 0.52 (BM25) for T5. These results suggest that BM25 serves as a reasonably effective proxy for relevance, particularly when explicit scores are unavailable, albeit with  slight performance degradation. Overall, the findings demonstrate that RankSHAP remains robust with implicit relevance measures, though explicit scores offer a tangible advantage, particularly in scenarios requiring finer-grained fidelity measures.
>
> We will explore opportunities to integrate this into the main paper after addressing all reviewer comments.

---

> ### Author Response · Authors · 2024-11-24
> **Other Questions**
>
> >Since the model can be black box, what if the model is itself self-contradictory or inconsistent? For example, a listwise ranking models that ranks documents A and B as A>B>C in the presence of C but as D>B>A in the presence of a document D. Would the explanations provided by RankSHAP in such a scenario inconsistent or unfaithful?
>
> Thank you for posing these insightful questions addressing potential challenges when RankSHAP is applied to scenarios involving self-contradictory or stochastic models.
> - Self-Contradictory or Inconsistent Models : In scenarios where the ranking model exhibits inconsistencies (e.g., ranking \( A > B > C \) in one context and \( D > B > A \) in another), RankSHAP faithfully reflects the model’s behavior as it is. This means that any inconsistencies in the model’s ranking logic will be captured in the feature attributions. A potential solution to that wouod be to introduce *consistency checks* in the RankSHAP pipeline to flag instances where the model exhibits contradictory behavior across subsets of documents.
> - Stochastic Rankings :   In case of stochastic rankings, RankSHAP can be adapted to compute *expected Shapley values* by sampling multiple permutations and averaging the feature attributions across repeated runs. This approach provides stable explanations that capture the probabilistic nature of the model. The randomness in rankings may introduce variability in the attributions. Reporting confidence intervals or variances in Shapley values helps users understand this uncertainty.
>
>
>
>
>
>
> >Can you explain why the fidelity scores are different for the two datasets? You mention that it is due to the difference in dataset sizes but I am not sure if it is clear if that is the reason.
>
> We appreciate your question and acknowledge that dataset size alone may not fully explain the difference in fidelity scores between Robust04 and MS MARCO. While the smaller size of Robust04 likely impacts the fine-tuning process, leading to less robust convergence, another contributing factor is the transferability of fine-tuned models. The neural ranking models in the Robust04 experiments were initially fine-tuned on MS MARCO, which is a much larger dataset with different query-document characteristics. This domain shift can result in suboptimal performance when applied to Robust04, impacting the fidelity scores. In addiiton, the complexity of queries and the relevance distribution within documents in the datasets can influence fidelity as well. We will clarify these points in the revised manuscript to ensure this distinction is more evident.
>
> > User study caveats: (a) Preconceived Notions: The authors observed that randomly generated feature attributions achieved a higher concordance score in the re-ordering task than expected based on their metric evaluation. This suggests that participants might have relied on pre-existing assumptions or biases about the topics, potentially influencing their judgments throughout the experiment. Is that a drawback of the setup that also impacts rest of the observations? (b) Subjectivity: The authors noted significant variance in the queries estimated by participants for the same document set and feature attributions. This probably highlights the inherent subjectivity in interpreting feature attributions and formulating queries, which can lead to diverse responses and impact the evaluation's reliability. Despite limitations, it is indeed on interesting study to include in the paper since progress in the field of interpretability requires human subjects to be involved.
>
> Thank you for your thoughtful observations regarding the user study caveats. We acknowledge that factors like preconceived notions and subjectivity introduce complexities in interpreting user study results. While these observations are beyond the scope of this work, they open up fascinating directions for future research. For instance, how can we design experiments to minimize the influence of participants' biases or preconceived notions about the topics? Additionally, what methodologies can be developed to quantify and account for subjectivity when evaluating feature attributions? Exploring these questions could enhance the reliability and robustness of user studies in interpretability research. We appreciate your recognition of the value of including such studies despite their inherent challenges and will consider these points in future work.

---

> ### Comment · Reviewer_URTs · 2024-12-02
>
> I have read the author's response to my questions. I will keep my score unchanged and do not have any outstanding questions.

---

> > ### Author Response · Authors · 2024-12-02
> >
> > We are deeply grateful for your thoughtful review and strong positive evaluation of our work. Your feedback played a crucial role in enhancing the quality of this paper.

---

### Author Response · Authors · 2024-11-27
**Author's Summary of Rebuttal Discussion**

We sincerely thank all reviewers for their thoughtful feedback and valuable suggestions. We are grateful that reviewers have appreciated our **robust and generalizable axiomatic framework** *(URTs, ExvH, UYR5)*, **thoughtful adaptation of the Shapley value for ranking** *(URTs, TGSB)*, **discussion of computational feasibility** *(URTs)*, **extensive experiments** *(URTs, ExvH)*, **thorough empirical evaluation** *(UYR5)* and the **user study, which adds practical value** *(URTs, ExvH, TGSB)*.

This is a summary of main concerns in the initial reviews and how we addressed them:
1. *"RankSHAP experiments with other value functions" (TGSB)* : We have extended our evaluation of RankSHAP by incorporating additional value functions that satisfy GREM, including MAP, CG, and DCG. **The performance of RankSHAP with various GREM-compliant value functions is detailed in Appendix I and summarized in Table 4.**

2. *"RankSHAP computation when explicit relevance labels are unavailable" (URTs,ExvH)* : We suggest using heuristic relevance functions like BM25 in such cases. We have explored the performance of RankSHAP when utilizing explicit relevance labels compared to heuristic-based relevance labels, such as those derived from BM25, in cases where explicit relevance labels are unavailable. **The comparison of RankSHAP’s performance using explicit versus heuristic relevance measures is presented in Appendix J and Table 5.**

These results have also been incorporated into the main paper where appropriate.

Importantly, the reviewers note that RankSHAP is a **thoughtful adaptation of the Shapley value for the ranking domain**, and that **defining new ranking-specific properties** enables **SHAP's applicability in ranking contexts** *(URTs,TGSB)*.

Updates :

1. Reviewer URTs has read our rebuttal, maintains their positive score and does not have any outstanding questions.

2. Reviewer ExvH is satisfied with our rebuttal and has increased their score following it.

---

### Meta-Review · Area_Chair_PJie · 2024-12-23

**Metareview:**

The paper extends the idea to using Shapley values for feature attributed to ranking tasks. In addition to the standard axioms of the original Shapley values, the proposed extension satisfies two axioms specific to ranking tasks. Further, the paper demonstrates empirically that their extension to Shapley values outperform existing methods in terms of accuracy and alignment with human intuition. All the reviewers appreciated the contribution and even though they highlighted a number of points for improvement, which the reviewers successfully addressed during the rebuttal period. In terms of overall score, all the reviewers are mildly positive or positive. As a consequence, I recommend acceptance.

**Additional Comments On Reviewer Discussion:**

The authors put a significant effort in addressing the reviewers' concerns in their rebuttal and they were able to persuade three reviewers to increase their overall score.

---

### Decision · Program_Chairs · 2025-01-22

Accept (Poster)